

# An AeroCom/AeroSat study: Intercomparison of Satellite AOD Datasets for Aerosol Model Evaluation

Nick Schutgens[1], Andrew M. Sayer[2,8], Andreas Heckel[3], Christina Hsu[4], Hiren Jethva[5], Gerrit de Leeuw[6], Peter J.T. Leonard[7], Robert C. Levy[4], Antti Lipponen[9], Alexei Lyapustin[10], Peter North[3], Thomas Popp[11], Caroline Poulson [12], Virginia Sawyer[13,4], Larisa Sogacheva[6], Gareth Thomas[14], Omar Torres[15], Yujie Wang[16], Stefan Kinne[17], Michael Schulz[18], and Philip Stier[19]

[1]Department of Earth Science, Vrije Universiteit Amsterdam, 1081 HV Amsterdam, the Netherlands
[2]Universities Space Research Association, Columbia, USA
[3]Department of Geography, Swansea University, Swansea, UK
[4]Climate and Radiation Laboratory, Earth Science Division, NASA Goddard Space Flight Center, Greenbelt, USA
[5]Universities Space Research Association-GESTAR, NASA Goddard Space Flight Center, Greenbelt, USA
[6]Finnish Meteorological Institute (FMI), Climate Research Programme, Helsinki, Finland
[7]ADNET Systems, Inc., Suite A100, 7515 Mission Drive, Lanham, MD 20706, US
[8]Ocean Ecology Laboratory, NASA-Goddard Space Flight Center (GSFC), Greenbelt, Maryland, USA
[9]Atmospheric Research Centre of Eastern Finland, Finnish Meteorological Institute, Kuopio, Finland.
[10]Laboratory for Atmospheres, NASA Goddard Space Flight Center (GSFC), Greenbelt, Maryland, USA
[11]German Aerospace Center (DLR), German Remote Sensing Data Center Atmosphere, Oberpfaffenhofen, Germany.
[12]School of Earth Atmosphere and Environment, Monash University, Australia
[13]Science Systems and Applications (SSAI), Lanham, Maryland, USA
[14]Remote Sensing group, Rutherford Appleton Laboratory, Harwell Campus, Didcot, Oxfordshire, UK.
[15]Atmospheric Chemistry and Dynamics Laboratory, NASA Goddard Space Flight Center (GSFC), Greenbelt, USA
[16]University of Maryland Baltimore County, Baltimore, Maryland, USA.
[17]Max-Planck-Institut für Meteorologie, Hamburg, Germany
[18]Norwegian Meteorological Institute, Blindern, Oslo,Norway
[19]Atmospheric, Oceanic and Planetary Physics, Department of Physics, University of Oxford, UK.

**Correspondence:** Nick Schutgens (n.a.j.schutgens@vu.nl)

**Abstract.** To better understand current uncertainties in the important observational constraint to climate models of AOD (Aerosol Optical Depth), we evaluate and intercompare fourteen satellite products, representing 9 different retrieval algorithm families using observations from 5 different sensors on 6 different platforms. The satellite products, super-observations consisting of $1^o \times 1^o$ daily aggregated retrievals drawn from the years 2006, 2008 and 2010, are evaluated with AERONET

5 (AErosol RObotic NETwork) and MAN (Maritime Aerosol Network) data. Results show that different products exhibit different regionally varying biases (both under- and overestimates) that may reach $\pm 50\%$, although a typical bias would be $15-25\%$ (depending on product). In addition to these biases, the products exhibit random errors that can be 1.6 to 3 times as large. Most products show similar performance, although there are a few exceptions with either larger biases or larger random errors. The intercomparison of satellite products extends this analysis and provides spatial context to it. In particular, we show that

10 aggregated satellite AOD agrees much better than the spatial coverage (often driven by cloud masks) within the $1^o \times 1^o$ grid cells. Up to 50% of the difference between satellite AOD is attributed to cloud contamination. The diversity in AOD products





shows clear spatial patterns and varies from 10% (parts of the ocean) to 100% (central Asia and Australia). More importantly, we show that the diversity may be used as an indication of AOD uncertainty, at least for the better performing products. This provides modellers with a global map of expected AOD uncertainty in satellite products, allows assessment of products away from AERONET sites, can provide guidance for future AERONET locations, and offers suggestions for product improvements.

We account for statistical and sampling noise in our analyses. Sampling noise, variations due to the evaluation of different subsets of the data, causes important changes in error metrics. The consequences of this noise term for product evaluation are discussed.

# 1   Introduction

Aerosol is an important component of the Earth's atmosphere that affects the planet's climate, the biosphere, and human health. Aerosol particles scatter and absorb sunlight as well as modify clouds. Anthropogenic aerosol changes the radiative balance and influences global warming (Angstrom, 1962; Twomey, 1974; Albrecht, 1989; Hansen et al., 1997; Lohmann and Feichter, 2005, 1997). Aerosol can transport soluble iron, phosphate and nitrate over long distances and so provide nutrients for the biosphere (Swap et al., 1992; Vink and Measures, 2001; McTainsh and Strong, 2007; Maher et al., 2010; Lequy et al., 2012) .

Finally, aerosol can penetrate deep into lungs and may carry toxins or serve as disease vectors (Dockery et al., 1993; Brunekreef and Holgate, 2002; Ezzati et al., 2002; Smith et al., 2009; Beelen et al., 2013; Ballester et al., 2013).

The most practical way to obtain observations on the global state of aerosol is through remote sensing observations from either polar orbiting or geostationary satellites (Kokhanovsky and de Leeuw, 2009; Lenoble et al., 2013; Dubovik et al., 2019). Unfortunately, that is a complex process as it requires a relatively weak aerosol signal to be distinguished from strong reflec-

tions by clouds and the surface. Even if cloud-free scenes are properly identified and surface reflectances properly accounted for, aerosols themselves come in many different sizes, shapes and compositions that affect their radiative properties. It is challenging to remotely sense aerosol, as this is essentially an under-constrained inversion using complex radiative transfer calculations.

Therefore it is no surprise that much effort has been spent on developing sensors for aerosol, the algorithms that work on

them and the evaluation of the resulting retrievals. Among the retrieved products, AOD (Aerosol Optical Depth) is the most common retrieval and the topic of this paper.

Intercomparison of a small number of satellite datasets probably goes back to a spirited discussion of the (dis)agreement between L2 MODIS (MODerate resolution Imaging Spectroradiometer) and MISR (Multi-angle Imaging SpectroRadiometer) AOD (Liu and Mishchenko, 2008; Mishchenko et al., 2009, 2010; Kahn et al., 2011). Not only did these studies show the value

in intercomparing satellite datasets (in part to compensate for the sparsity of AERONET sites), but also the various challenges in doing so. Evaluation and intercomparison of satellite AOD products is difficult for a number of reasons: the data exists





in different formats, for different time periods that may overlap only partially, computational requirements (especially for L2 data) are large, and the data usually come in different spatio-temporal grids. In addition, data has often been filtered in different ways and aggregates produced differently. Listing all papers that intercompare two or three satellite datasets would probably not be accepted by the editors of this journal, so in Table 1, we constrain ourselves to publications with at least 5 different

datasets.

Most of the papers in Table 1 quantify only global biases for daily or monthly data. Half of them use monthly satellite data, potentially introducing significant temporal representation errors (Schutgens et al., 2016b) in their analysis. Seldom is the spatial representativity of AERONET sites accounted for (Schutgens et al., 2016a) although most studies do exclude mountain sites. As a result both the evaluations with AERONET and the satellite product intercomparisons are no apples-to-

apples comparisons. Finally, most studies do not systematically address (statistical or sampling) noise issues inherent in their analysis.

In this paper, we will assess spatially varying (as opposed to global) biases in multi-year averaged satellite AOD (appropriate for model evaluations). As truth references AERONET and MAN data will be used. The analysis uses only AERONET sites with high spatial representativity and collocates all data within a few hours, greatly reducing representation errors. Throughout

a bootstrapping method is used to assess statistical noise in the analysis. Sampling issues (e.g. the sparsity of AERONET sites) are addressed through e.g. a pair-wise satellite intercomparison.

This paper is the result of discussions in the AeroCom (AEROsol Comparisons between Observations and Models, https://aerocom.met.no) and AeroSat (International Satellite Aerosol Science Network, https://aero-sat.org) communities. Both are grass-roots communities, the first organised around aerosol modellers, and the second around retrieval groups. They meet

every year to discuss common issues in the field of aerosol studies.

The structure of the paper is as follows. The remote sensing products are described in Section. 2 and the methodology to collocate them in space and time in Section 3. Section 4 describes screening procedures for representative AERONET sites and establishes the robustness of our collocation procedure. Section 5 evaluates the satellite products individually against AERONET and MAN, at daily and multi-year timescales. An intercomparison of pairs of satellite products is presented in

Section 6. A combined evaluation and intercomparison of the products is made in Section 7. More importantly, the diversity amongst satellite products is discussed and interpreted. A summary can be found in Section 8.

## 2   Remote sensing data

Original satellite L2 data were aggregated unto a regular spatio-temporal grid with spatio-temporal grid-boxes of $1^o \times 1^o \times 30^{\min}$. The resulting super-observations ($1^o \times 1^o \times 30^{\min}$ aggregates) are more representative of global model grid-boxes ($\sim$

$1^o - 3^o$ in size) while allowing accurate temporal collocation with other datasets. At the same time, the use of super-observations significantly reduces data amount without much loss of information (at the scale of global model grid-boxes). A list of products used in this paper is given in Table 2. A colour legend to the different products can be found in Fig. 1.



The actual aggregation consists of finding all L2 retrievals that belong to a spatio-temporal grid-box and calculating an arithmetic average. Note that different averages might be used (e.g. geometric, see also Levy et al. (2009); Sayer and Knobelspiesse (2019)) but a modelling study (Schutgens et al., 2017), backed up by limited sensitivity studies using the present datasets, suggest relatively little impact in the intercomparison of datasets.

The main data is AOD at 550 nm, the wavelength at which models typically provide AOD. If AOD was not retrieved at this wavelength, it was interpolated (or extrapolated) from nearby wavelengths. In addition, the standard deviation over the original L2 retrievals and a retrieval error estimate per super-observation were included. If possible, super-observations at multiple wavelengths were obtained, usually 440 and 870 nm.

In addition, the number of L2 retrievals used per super-observation, as well the average pixel size for these L2 retrievals were
included. For some products (e.g. MODIS), this physical pixel size will vary as the view angle changes across the imager's field-of-view. In that case, actual pixel footprints can be difficult to calculate due to the Earth's curvature (Sayer, 2015) and only estimates were provided. Other products (MAIAC and those from AATSR) are based on regridded radiance data and use a fixed pixel size. The combination of number of retrievals and average pixel size can be used to estimate the spatial coverage: the fraction of a $1^o \times 1^o$ grid-box covered by retrievals (at a particular time) per super-observation. This spatial coverage would
ideally be 100% but in practice is normally smaller for several reasons: the imager's field-of-view may miss part of the $1^o \times 1^o$ grid-box; sun glint, snow, desert surface or clouds, may prevent retrievals; or retrievals may fail. We will provide evidence that cloud masking is the dominant factor, see also Zhao et al. (2013), which suggests that spatial coverage might be interpreted as an estimate of the complement to cloud fraction.

All products were provided globally for three years (2006, 2008 and 2010, which includes two years used in AEROCOM
control studies). Many products only provided data over land. Seven datasets belong to sensors that have an equatorial crossing time in the morning, and another seven belong to sensors that have an equatorial crossing time in the afternoon.

AERONET DirectSun L2.0 V3 and MAN L2.0 data were downloaded from `https://aeronet.gsfc.nasa.gov` and aggregated per site by averaging over 30 minutes. MAN aggregates were assigned averaged longitude and latitudes for those 30 minutes.

The entire satellite dataset requires 14GB of storage. All data are stored in netCDF format.

## 3   Collocation & analysis methodology

To be able to evaluate and intercompare the remote sensing datasets, they will need to be collocated in time and space to reduce representation errors (Colarco et al., 2014; Schutgens et al., 2016b, 2017). This is achieved by only retaining data from multiple datasets if they occur within the same spatio-temporal window. In practice this collocation is another aggregation
(performed for each dataset individually) to a spatio-temporal grid with slightly coarser temporal resolution (1 or 3 hours, the spatial grid-box size remains $1^o \times 1^o$), followed by a masking operation that discards data at times and locations not present in all datasets. More information on the procedure can be found in Watson-Parris et al. (2016) which also introduces a powerful





and flexible command-line tool and Python library called Community Intercomparison Suite (CIS, `www.cistools.net`, last accessed on December 20, 2019), for such operations.

Station data, whether AERONET or MAN, is allocated to whichever grid-box they fall in. Point observations will always suffer from representativeness issues (Sayer et al., 2010; Virtanen et al., 2018; Schutgens et al., 2016a), but the representativity

of AERONET sites for $1^o \times 1^o$ grid-boxes is fairly well understood (Schutgens, 2019), see also Section 4.

A satellite product will contribute at most a single super-observation to any spatio-temporal grid-box of this slightly coarser grid (as satellite revisit times are well in excess of the grid's resolution). However, AERONET data (already aggregated over $30^{min}$, see previous section) may contribute up to 2 super-observations per hour and even 6 super-observations per 3 hours (in which case they are averaged).

As the super-observations are on a regular spatio-temporal grid and collocation requires further aggregation to another regular but coarser, grid, the whole procedure is very fast. It is possible to collocate all 7 products from afternoon platforms over three years using an IDL (Interactive Data Language) code (that served as a prototype for CIS) and a single processing core in just 30 minutes. This greatly facilitates sensitivity studies.

After spatio-temporally collocating two or more datasets, the data may be further averaged in space and/or time for analysis

purposes. For instance, by averaging in time it becomes possible to make global maps; by averaging in space it becomes possible to construct regional time-series. During e.g. temporal averaging no attempt is made to account for varying data availability throughout the months and years, in order to keep temporal representation errors at a minimum (Schutgens et al., 2016b).

During the evaluation of products with AERONET, a distinction will be made between either land or ocean grid-boxes in

the common grid. A high resolution land mask was used to determine which $1^o \times 1^o$ grid-box contained at most 30% land (designated an ocean box) or water (designated a land box). Most ocean boxes with observations will be in coastal regions, with some over isolated islands.

### 3.1 Taylor diagrams

A suitable graphic for displaying multiple datasets' correspondence with a reference dataset ('truth'), is provided by the Taylor

diagram (Taylor, 2001). In this polar plot, each data point $(r, \phi)$ shows basic statistical metrics for an entire dataset. The distance from the origin ($r$) represents the internal variability (standard deviation) in the dataset. The angle $90^o - \phi$ through which the data point is rotated away from the vertical axis represents the correlation with the reference dataset, which is conceptually located at the horizontal axis at radius 1 (i.e. every distance is normalized to the internal variability of the reference dataset). It can be shown (Taylor, 2001) that the distance between the point $(r, \phi)$ and this reference data point at $(1, 0)$ is a measure of

the (normalized) Root Mean Square Error (RMSE, unbiased). A line extending from the data point can be used to show the bias versus the reference dataset (positive for pointing clock-wise), again normalized to the internal variability in the reference dataset. The distance from the end of this line to the reference data point is a measure of the (normalized) Root Mean Square Difference (RMSD, no correction for bias).



## 3.2 Uncertainty analysis using bootstrapping

Our estimates of error metrics are inherently uncertain due to finite sampling. If the sampled error distribution is sufficiently similar to the underlying true error distribution, bootstrapping (Efron, 1979) can be used to assess uncertainties in e.g. biases or correlations due to finite sample size. Bootstrapping uses the sampled distribution to generate a large number of synthetic samples by random draws *with replacement* from the observed distribution. For each of these synthetic samples, a bias etc. can then be calculated and an uncertainty can be calculated as e.g. the standard deviation over all these biases. Bootstrapping has been shown to be reliable even for relatively small sample sizes. In this study, the uncertainty bars in many figures were generated by bootstrap analysis.

If the sampled error distribution is different from the true error distribution, bootstrapping will likely underestimate uncertainties. Sampled error distributions may be different from the true error distribution because the act of collocating satellite and AERONET data favours certain conditions. For example, the effective combination of two cloud screening algorithms (one for the satellite product, the other for AERONET) may favour clear sky conditions and limit sampling of errors in case of cloud contamination. This uncertainty due to sampling is harder to assess but we attempt to address it by comparing evaluations for different combinations of collocated satellite products.

## 3.3 Error metrics for evaluation

For most of this study we will focus on the usual global error statistics (bias, RMSD, Pearson correlation), treating all data as independent. Global statistics may be dominated by a few sites with many collocations, which will skew results. We also performed analyses on regional scales but they will not be shown. Instead we also show novel error metrics: the bias (sign-less) and the correlation per site, averaged over all sites. This prevents a few sites from skewing the error statistics. Only sites with at least 32 collocations will be used in this novel analysis.

## 4 Selection of AERONET sites and collocation criteria

Not all AERONET sites are equally suited for the evaluation of satellite data: both maintenance quality and spatial representativity vary by site. Kinne et al. (2013) provides a subjective ranking of all sites (before 2009) based on their general level of maintenance and spatial representativity. The ranking is based on personal knowledge of the sites and is mostly qualitative. Schutgens (2019) provides an objective ranking for all sites (for all years) based on spatial representativity alone. This ranking is based on a high resolution modelling study and quantitative. While there is substantial overlap in their rankings for spatial representativity there are also differences. Table 3 describes the AERONET site selections used in this paper.

Figure 2 shows a comparison of global biases, correlations and RMS differences of satellite super-observations when evaluated with AERONET using either all sites or the Kinne subset (satellite products were individually collocated with AERONET within 1 hour). Using the Kinne subset significantly lowers the total number of available collocated measurements but also



slightly increases correlations and decreases RMS differences. No systematic change in global bias is discernible. Averages for these metrics over all products are given in Table 4.

According to this table, using the Schutgens subset of AERONET sites yields a small improvement in correlation (over the Kinne subset) yet allows for more collocated observations (the reduction in product averaged global bias is likely the result

of balancing errors, and deemed not meaningful). As we later want to evaluate satellite products at individual sites, we will continue to use the Kinne subset which considers not only spatial representativity but also site maintenance.

To verify the Kinne selection, we used the satellite products themselves. First we selected sites that provide collocated observations with individual satellite products for at least 8 months across at least 2 years (rather arbitrary but it seemed not to matter much). Next we calculated for each satellite product a bias and correlation with respect to each site. Finally, we

computed for each site, the maximum correlation and minimum relative (sign-less) bias across all products. The idea here is that if *all* satellite products perform very poorly over a site, it may just be that the site itself is unsuited (due to maintenance or spatial representativity issues that were not flagged up in Kinne et al. (2013). Results are shown in Figure 3. Clearly a few sites stand out: Canberra, Crozet Island, Amsterdam Island and Tinga Tingana. For Canberra and Crozet Island, products have significantly lower correlation than for the majority of sites. For Tinga Tingana, products significantly overestimate AOD

compared to the majority of sites. The Amsterdam Island site exhibits both poor correlations and large biases. Removing these from the Kinne selection has however only a small impact on global statistics, see Table 4. We will nevertheless use this pruned selection in the remainder of this paper. Note that only for a minority of sites (10%) all satellite products will either over-estimate or under-estimate AOD. For most sites, the products form an ensemble of AOD values that straddle the AERONET value. In this analysis, the satellite products are *not* on the same common *temporal* grid.

To further confirm the suitability of the remaining AERONET sites, we present the following analysis. The difference between a satellite $\tau_{\mathrm{sat}}$ and AERONET $\tau_{\mathrm{A}}$ super-observation AOD can be understood as the sum of observation errors in both products and a representation error $\epsilon_{\mathrm{rep}}$ (the latter accounts for the site's suitability to represent a $1^o \times 1^o$ grid-box) :

$$\tau_{\mathrm{sat}} - \tau_{\mathrm{A}} = \epsilon_{\mathrm{sat}} + \epsilon_{\mathrm{A}} + \epsilon_{\mathrm{rep}}. \tag{1}$$

If we assume these errors are uncorrelated and have a Gaussian distribution, it follows that

$$\epsilon_{\mathrm{norm}} = \frac{\tau_{\mathrm{sat}} - \tau_{\mathrm{A}}}{\sqrt{\sigma_{\mathrm{A}}^2 + \sigma_{\mathrm{rep}}^2}} > \frac{\tau_{\mathrm{sat}} - \tau_{\mathrm{A}}}{\sqrt{\sigma_{\mathrm{sat}}^2 + \sigma_{\mathrm{A}}^2 + \sigma_{\mathrm{rep}}^2}}, \tag{2}$$

should have a Gaussian distribution with standard deviation larger than 1. Here $\sigma_{\mathrm{sat}}$ is the unknown uncertainty in satellite observations, $\sigma_{\mathrm{A}} = 0.01$ the known uncertainty in AERONET AOD (Eck et al., 1999; Schmid et al., 1999) and $\sigma_{\mathrm{rep}}$ the uncertainty due to representation errors. The latter can be estimated as follows: if satellite observation errors across a $1^o \times 1^o$ grid-box are mostly constant, the standard deviation $\sigma_{\tau}$ of L2 AOD over this grid-box may be taken as an estimate of the

representation uncertainty. Figure 4 shows this normalized error distribution for two satellite products although results are similar for others. As the distribution is significantly wider than a Gaussian with standard deviation of 1, it appears that the observation errors are twice as large as the representation errors. See Sayer et al. (2019a) for a different application of a very similar statistical test.



Finally, we investigate the impact of the temporal collocation criterium and the required minimum number of AERONET super-observations on product evaluation, see Table 5. It turns out that changing this number has only a small impact on the product evaluation, see also Fig. 5. Intriguingly, the highest product correlations and lowest RMS differences with AERONET are found for a collocation requirement of 3 hours and at least 5 AERONET measurements and not for a tighter constraint of 1

5 hour. In Schutgens et al. (2017) it was shown that point measurements become more spatially representative for a larger area by temporal averaging. It was estimated that a $1^o$ grid-box was best represented by a point observation if its measurements were averaged over 4 hours. However, it is also possible that requiring at least 5 AERONET observations selects for very clear grid-boxes (i.e. no cloud contamination in the satellite products). Given the substantial reduction in available collocated observations, we decided to require only a single AERONET super-observation for succesful collocation.

We also considered the impact of these choices on regional evaluations. Broadly, similar conclusions can be drawn although the analysis can become rather noisy due to smaller sample sizes.

## 5 Evaluation of individual satellite products

In this section we will evaluate individual satellite products with either AERONET or MAN observations. In both cases, the data were collocated within one hour.

In Fig. 6 we see the evaluation with AERONET, using Taylor diagrams (Taylor, 2001), see also Sect. 3.1. Over land the MODIS algorithms generally do very well, showing high correlations although biases and standard deviations can be quite different. The same algorithm applied to either Aqua or Terra yields very similar results in the Taylor diagram. The exception is a relatively high bias for Terra-DT. The AATSR products generally have lower correlations than the MODIS products although AATSR-ADV comes close. It is interesting to compare three products (Aqua-DB, SeaWiFS and AVHRR) that use a

similar algorithm but with different amounts of spectral information. MODIS and SeaWiFS perform very similar but AVHRR shows much lower correlation. Some products globally over-estimate AOD at the AERONET sites whilst others under-estimate it (see also Fig. 3). As the data count over land is high, statistical noise in these statistics are negligible, as can also be seen in Fig. 2 which is dominated by land sites.

Over ocean, the message is more mixed. The AATSR products do relatively better, while Terra/Aqua-DB seems to be

slightly outperformed by AVHRR and SeaWiFS (note that Terra/Aqua-DB and BAR only retrieve data over land and the "over ocean" analysis is confined to coastal regions). Over ocean no products significantly underestimate global AOD although a few (e.g. SeaWiFS and Aqua-MAIAC) have small negative biases. Several products significantly over-estimate global AOD (e.g. DarkTarget, OMAERUV and AATSR-ORAC). The data count for over ocean evaluation is not very high and consequently statistical noise in this analysis is larger than over land. A sensitivity study using bootstrapping (see Sect. 3.2) nevertheless

suggests these results are quite robust. They are also partially supported by product evaluation with MAN, in Fig. 7: the AATSR products do better than the other products but it is clearly possible for the products to either over- or underestimate AOD globally. The data count for MAN evaluation is low and statistical noise is large, see Fig. 8. For e.g. OMAERUV, the uncertainty range suggests this could be either one of the worst or best performing products.



If we split each product's data into two equally sized subsets depending on the collocated AERONET AOD (median AOD $\sim 0.12$), it becomes obvious that the satellite products have much lower skill at low AOD, see Fig. 9. They correlate much worse with AERONET, show much higher internal variability than AERONET and exhibit relatively larger biases than at high AOD Note that biases at low AOD are all positive while at high AOD they are negative (exception: Terra-DT).

In Fig. 10, we consider the impact of spatial coverage on product evaluation. As minimum spatial coverage increases, the correlation with AERONET increases while the bias decreases. If spatial coverage is mostly determined by cloud screening, it seems reasonable that cloud contamination of AOD retrievals increases as spatial coverage decreases. This would lead to the observed behaviour of biases and correlations. In contrast, it is hard to use other factors determining spatial coverage (sun-glint, surface albedo, failed retrievals) to explain this. We also note that the change with spatial coverage is quite dramatic for some

products (AVHRR, OMAERUV, AATSR-ORAC) while for others it is rather small. It is not surprising that some products will have better cloud masking than others (depending e.g. on pixel sizes). It appears hard to determine a threshold value beyond which there is no substantial change in all the metrics in the majority of products, so we continued to use all data.

    The impact of temporal averaging on product differences is shown in Fig 11. The "daily" column shows distributions for individual super-observations, while the "3-years" column shows 3-year averages (averaged per site). Temporal averaging

significantly reduces differences, e.g. the typical AVHRR difference decreases almost threefold from 0.077 to 0.027. In contrast, the typical difference for OMAERUV decreases only from 0.094 to 0.059, a factor of 1.6. It seems OMAERUV exhibits larger biases than AVHRR which has rather large random differences. As noted before, the major part of the daily difference is due to observation errors while a smaller part is due to representation errors. Previous analyses (Schutgens et al., 2016a, 2017; Schutgens, 2019) and our selection of AERONET sites, suggest the 3-year average AOD difference will only have a small

contribution from representation errors. After that amount of averaging, statistical analysis suggests that the typical 3-year differences may be interpreted as biases, i.e. the typical multi-year bias *per site* in Aqua-DT is 0.029. All products exhibit both positive and negative biases across the AERONET network. The global mean bias of a product (the big black dot in Fig. 11) is usually much smaller than the bias at any site and results from balancing errors across the network.

    Note that the Terra-DT bias is significantly larger than Aqua-DT's bias in Fig. 11. Levy et al. (2018) discuss a systematic

difference between Terra and Aqua DarkTarget AOD which they attributed to remaining retrieval issues.

    Another way to evaluate the products is presented in Fig. 12, which shows the average correlation between any product and an AERONET site versus the average relative (sign-less) bias with an AERONET site. This analysis is very different from the Taylor analysis presented earlier, where both correlation and bias were calculated across the entire dataset, instead of per site and then averaged across all sites. Figure 12 suggests that product biases per site are typically some 20%. The relative

performance of the products shows significant differences with the earlier Taylor analysis: AATSR-SU is now one of the top performers while Terra/Aqua-DB show $1.3\times$ larger biases than either Terra/Aqua-BAR or AATSR-SU (in the Taylor analysis Terra/Aqua-DB has one of the smallest global biases).

    Both in Fig. 11 and 12, we have considered only AERONET sites that provide a minimum of 32 collocated observations. Although each product was individually collocated with AERONET, only those sites that are common across all product

collocations were retained for analysis.





A more detailed look at each product and its evaluation against AERONET is provided in Figures 13, 14, 15 and 16. Shown are a scatter plot of (daily) collocated super-observations vs AERONET; the impact of spatial coverage on the difference between satellite and AERONET AOD, a global map of the 3-year averaged product AOD; and a global map of the difference of 3-year averaged product AOD with AERONET (again, using only sites with 32 or more collocations).

The scatterplots typically show good agreement with AERONET: correlations vary from 0.73 to 0.89 with regression slopes of 0.99 possible (mean and standard deviation refer to the difference with AERONET). The impact of spatial coverage on the differences with AERONET are consistent for all products and relatively muted, as also seen in the right panel of Fig. 10. The global maps of AOD show first of all the extent of the product: Terra/Aqua-DB, MAIAC and BAR provide no significant coverage of the oceans while OMAERUV mostly seems to cover the large outflows over ocean. MAIAC is currently miss-

ing a sizeable portion of Siberia. Terra/Aqua-DT & BAR, AATSR-FMI-ADV and to a lesser degree AVHRR do not retrieve over the desert regions in Northern Africa and the Middle East. Terra/Aqua-DT, by the way, sometimes produces negative AOD leading to e.g. very low values for averaged AOD over Australia. In the global maps of 3-year averaged differences with AERONET, land sites are shown in circles, ocean sites in squares and the remainder as diamonds. These maps show distinct spatial patterns: e.g. Aqua-DT mostly overestimates AOD in the northern hemisphere and underestimates in the south-

ern hemisphere; OMEARUV overestimates everywhere except in the African greenbelt and south-east Asia; MAIAC mostly underestimates AOD (MAIAC MODIS C6 lacks seasonal dependence of aerosol models, which leads to an underestimation during the biomass-burning or dust seasons with high AOD. This will be corrected in C6.1). Regional patterns can also be seen, e.g. several products overestimate AOD in the eastern continental USA and underestimate it in the west.

## 6  Pair-wise intercomparison of the satellite datasets

In this section, we will intercompare the various satellite products by collocating them pair-wise within 1 hour. Our analysis will be split between products for either morning or afternoon platforms as this usually leads to a large amount of collocated data with an almost global distribution. However, even products from e.g. Terra and Aqua will provide some collocated data (at high northern latitudes) and will be discussed as well.

The difference in 3-yearly AOD is shown in Fig. 17 and Fig. 18 for morning and afternoon satellites respectively. We see

that the majority of collocated products provides only data over land. AOD differences behave very smoothly over ocean but show a lot of spatial variation over land. AOD differences can be significant and exceed $50\%$. Over ocean, the difference is longitudinally fairly homogenous with a clear latitudinal dependence. Over land, regional variability often tracks land features: the Rocky Mountains and Andes, the Sahara and African greenbelt can all be easily identified. That suggests albedo treatment as a driver of product difference. What is remarkable is the relatively large spatial scale involved. This analysis confirms the

one in the previous Section where spatial patterns in AOD bias against AERONET were discussed and extends it with more detail. The contrasts in the differences over land and neighbouring ocean (e.g. African outflow for Terra-DT with AATSR-SU or AATSR-ADV, or Aqua-DT with OMAERUV, or AVHRR with SeaWiFS) may likewise be driven by albedo treatment. The





OMAERUV product consistently estimates higher AOD than all other products, with the possible exception of areas with known absorbing aerosol.

Products retrieved using the same retrieval scheme but observations from different platforms can be intercompared as well (MODIS on Aqua & Terra). Collocations are now limited to a fairly narrow latitudinal belt near the North pole, see Fig. 19.

The differences in AOD appear much more muted, suggesting that algorithms are the major driver of product difference, not differences in orbital overpass times or issues with sensor calibration. This is further supported by the difference amongst e.g. AATSR products which employ different algorithms but the same measurements (Fig. 17). The three products based on the DeepBlue algorithm (Aqua-DB, AVHRR and SeaWiFS) suggest that already small algorithmic differences can yield significant differences.

Correlations of the collocated pairs of AOD super-observations were also considered, see Fig. 20. The products derived from Aqua and Terra MODIS measurements tend to correlate well, with lesser correlation amongst the AATSR products. Highest correlation is found for products using the same algorithm and similar sensor but a different platform (Terra/Aqua). The very low correlations for AATSR-ORAC with Aqua products stand out but no explanation was found. Here again only collocations over high Northern latitudes are available.

Fig. 21 shows scatter plots of AOD and spatial coverage for selected collocated products. It is obvious that the agreement in AOD is far greater than the agreement in spatial coverage. Only when we consider collocated products for the same algorithm from different satellites, can remarkable agreement be found (e.g. Terra/Aqua MAIAC). For different products using the same sensor, spatial coverage can differ greatly even though the observed scene is the same. Figure 22 shows the low correlations for spatial coverage for all collocated product pairs. Even though the products apparently identify different parts of a $1^o \times 1^o$

grid-box as suitable for aerosol retrieval, they still agree quite well on aggregated AOD.

The impact of spatial coverage on AOD agreement for selected collocated products is shown in Fig. 23. AOD agrees better when the spatial coverage is high and this is more pronounced in the wings of the difference distributions ("outliers"). If spatial coverage is, to first approximation, the complement of cloud fraction in a $1^o \times 1^o$ grid-box, it may be expected that higher spatial coverage correlates with less cloud contamination of AOD. Especially when the same algorithm is used (here DeepBlue), it is

hard to see what can differ between Aqua and Terra observations less than a few hour apart that can affect spatial coverage, except for cloud cover.

Fortunately, the impact on AOD is not that large: Fig. 24 shows the ratio of mean sign-less difference in AOD for spatial coverages of $90 - 100\%$ to $0 - 10\%$. Typically this ratio is a factor of 0.57. In other words,at low spatial coverage 50% of the difference may be due to cloud contamination. A similar weak dependence on AOD evaluation was seen in Figures 13, 14, 15

and 16. One possible explanation is that aggregation into super-observations has a beneficial impact by tempering retrieval errors from cloud contamination.



## 7 Intercomparison and evaluation of collocated morning or afternoon products

In this section, we will perform an apples-to-apples comparison of the satellite products, collocating either all morning or all afternoon products together. To ensure sufficient numbers of collocated data, the temporal collocation criterium was widened to 3 hours. Even so, a significant reduction in data amount results from collocating so many datasets. If we include AERONET in the collocation, the total count will go down from $\sim 28,000$ to about 4000 collocated cases.

The resulting Taylor diagram is shown in Fig. 25 and can be compared to Fig. 6. The Terra products show reduced correlation, now almost on par with the AATSR products. The Aqua and Terra products are not collocated together but, in contrast to Fig. 6, are clearly separated in the Taylor diagram. Also, the majority of datasets have negative biases with respect to AERONET. A more in-depth comparison, is shown in Fig. 26. RMSD shows the most conspicuous changes: across the board RMSDs for the simultaneous collocation of 7 satellite products with AERONET are much smaller. Global biases are shifted towards negative values: e.g. OMAERUV now has a much smaller bias, while Aqua-DT has a much larger negative bias. Correlations are unaffected except for the Terra and the AATSR-SU products. In all cases, the uncertainty ranges suggest that the differences are statistically significant.

Both evaluations in Fig. 26 are valid in their own right. The evaluation of individual products with AERONET yields large amounts of data, while the simultaneous collocation of multiple morning or afternoon products allows proper intercomparison, without the added uncertainty due to different spatio-temporal sampling. Depending on one's point-of-view it's possible to say that either results are not very different (considering all products, the relative performance of datasets does not change much) or quite different (considering the best performing products, significant changes are visible).

The simultaneous collocation of multiple products yields a subset of the collocated data that were studied in Section 5, although for every product the subset from the 'original' is different. Unfortunately, we have not been able to explain the different evaluation results. Due to the different collocation criteria, there are differences in the mean spatial coverage of the super-observations, in the relative number of collocations per AERONET site and per year. How this affects each product differs and no systematic variation was found to help explain results. Ultimately this is testament to the complex influence of observational sampling.

Collocating either the morning or afternoon products *without* AERONET allows us to study diversity between these datasets on a global scale. Relative diversity is here defined as the relative spread (standard deviation divided by mean) calculated at each grid-box from the 3-year averages AOD of 7 (collocated) products, see Fig. 27. Here we have used all 7 morning or afternoon products over most of the land. Over ocean, the major desert regions and Siberia not all products provided data and only a subset was used. Over ocean, only Terra-DT and the three AATSR products or Aqua-DT, AVHRR and SeaWiFS were used. Over the desert regions (outlined in blue) , only Terra-DB, Terra-MAIAC, AATSR-ORAC and AATSR-SU or Aqua-DB, Aqua-MAIAC, SeaWiFS and OMAERUV were used. Over Siberia (outlined in blue), no data were present for MAIAC.

Diversity is generally lowest over ocean, never reaching over 30% while over land values of 100% are possible. Over ocean, diversity is lowest for the afternoon products, presumably because only 3 products contribute (Aqua-DT, SeaWiFS and AVHRR) and two (SeaWiFS and AVHRR) use a similar algorithm (SOAR). The spatial distribution of diversity is fairly





smooth over ocean, in contrast to land where one sees a lot of structure. This was also seen in the intercomparison of satellite products in Sect. 6. For an earlier study of satellite AOD diversity, see Chin et al. (2014) in which a different definition of diversity, a different (and smaller) set of satellite products, and a different (sub-optimal) collocation procedure lead to rather different magnitudes and spatial patterns for diversity. In contrast, the diversity presented in Sogacheva et al. (2019), while

using a different definition and a different (sub-optimal) collocation procedure agrees more with the one presented in Fig. 27. There is substantial overlap in the satellite products used here and in Sogacheva et al. (2019), see Tables 1 and 2.

Also shown is the average correlation, i.e. de average of the correlation between all possible pairs of collocated products. Over the deep ocean (e.g. southern hemisphere Pacific ocean) correlations are low. It seems that only in outflow regions (e.g. Amazonian outflow, South African outflow, outflow from Sahara and African savannah, Asian outflow) the products will

strongly agree in their temporal signal over ocean. This suggests that the correlation depends on the strength of the AOD signal (see also Sect. 5 and Fig. 9). Over land, the correlation shows more variation. Interestingly, the correlation is high when the diversity is low and vice versa: e.g. Australia shows high diversity in 3-year mean AOD and very low correlation between individual AOD. This anti-correlation suggests that the same factor(s) that cause errors in 3-year averages also cause random errors in individual AOD.

The above results are pretty robust. E.g. by excluding OMAERUV (arguably the product with the largest errors due to its large pixel sizes and interpolation from UV wavelengths) from this analysis, the afternoon diversity over land looks even more like the morning diversity. Diversity maps for two other collocations (all AATSR products or all Aqua products) are shown in Fig. 28. The Aqua maps looks similar to before, but diversity is more muted for the AATSR products (but notice the same spatial patterns).

Diversity is an ensemble property of 7 collocated products and can be interpreted based on other ensemble properties: the mean AOD and the relative spread in spatial coverage, see Fig. 29. We interpret the mean AOD as an indication of signal-to-noise in the satellite retrievals, and the spread in the spatial coverage as uncertainty in cloud masking. We see that the diversity goes down when the mean AOD increases, and goes up when the uncertainty in cloud masking increases. This is as one would expect. Notice that, for the majority of locations, the actual diversity varies only from $\sim 20\%$ to $\sim 50\%$, e.g. no more than a

factor 2.5.

Diversity turns out to be more than just the spread across multiple satellite products. The absolute diversity $\delta$ in the satellite AOD can actually be interpreted as the uncertainty $\bar{\sigma}_{\mathrm{sat}}$ in multi-year averaged satellite AOD, at least in a statistical sense. Taking the 3-year averaged differences between a satellite and AERONET AOD (per site) from Sect 5, and dividing them by the diversity in the satellite ensemble (at that site), these normalized errors

$$\bar{\epsilon}_{\mathrm{norm}} = \frac{\bar{\tau}_{\mathrm{sat}} - \bar{\tau}_{\mathrm{A}}}{\delta} \tag{3}$$

exhibit Gaussian distributions with standard deviations close to 1, see Fig. 30. We assume that in 3-year averages, both AERONET observation errors and representation errors are negligible. Hence, $\delta \approx \bar{\sigma}_{\mathrm{sat}}$. To put it differently the product multi-year error can be statistically modelled as a random draw from a distribution with the absolute diversity as standard deviation. This works very well for Aqua-DT, DB, BAR, SeaWiFS and AATSR-SU products. It works less for Aqua-MAIAC which





shows a global bias (identified before, see Sect. 5) but still has a normalized error with standard deviation close to 1. The Terra and AVHRR products show larger spread in the normalized error, while AATSR-ORAC and OMAERUV show significantly larger spread. It seems that the products that do better in the evaluation (Fig. 6 and 25) have errors that behave according to the diversity.

The conclusion that, in the current satellite ensemble, satellite AOD uncertainty may be modelled from satellite AOD diversity is probably the most important find of this study and allows for several useful applications which will be discussed in Sect. 8.

## 8    Summary

A detailed evaluation and intercomparison of 14 different satellite products of AOD is performed. Compared to previous
studies of this kind, this one includes more (diverse) products and considers longer time periods, as well as of course more recent satellite retrieval products. Unlike previous studies it explicitly addresses the issue of uncertainty due to either statistical noise or sampling differences in datasets. While satellite products are assessed at both daily and multi-year time-scales, the purpose of this study is to understand satellite AOD uncertainty in the context of model evaluation. In practice this means $1^o \times 1^o$ aggregates (or super-observations) of the original retrievals are evaluated for their multi-year bias.

The 14 satellite products include retrievals from MODIS (Terra/Aqua), AATSR (ENVISAT), AVHRR (noaa18), SeaWiFS (SeaStar) and OMI (Aura). Two other products, based on POLDER (PARASOL) are part of the database but were not included in the current paper. They will be reported on in a follow-up paper. Yet two other products, MISR (Multi-angle Imaging SpectroRadiometer) and VIIRS (Visible Infrared Imaging Radiometer Suite), are not part of the current AEROCOM/AEROSAT study. MISR because the product was in the middle of an update cycle, and VIRSS because it was only launched in 2011. For
MODIS and AATSR, four resp. three different retrieval algorithms were used The over-land products from AVHRR, SeaWiFS and one MODIS product use variations of the same algorithm (DeepBlue).

The evaluation is made with AERONET and MAN observations. Only AERONET sites with good spatial representativity and maintenance records were selected, based on a previously published list by Kinne et al. (2013). The suitability of these sites was further assessed by "evaluating" them against the ensemble of satellite products which lead to the identification of four sites
that show substantially different AOD than any satellite dataset. Whether these sites are unsuitable to satellite evaluation or all products retrieve AOD poorly over those sites is an open question but we removed them from our selection of AERONET sites. Lastly we used the satellite observations themselves to confirm that representation errors, while not negligible, are a minor contribution to the difference between satellite ($1^o \times 1^o$ aggregates) and AERONET AOD.

For evaluation and intercomparison purposes, different data products were collocated within a few hours. Sensitivity studies
show this to provide a good trade-off between accuracy and data amount. We make extensive use of bootstrapping to assess the uncertainty ranges in our error metrics due to statistical noise. We try to address uncertainty due to the spatial sparsity of AERONET and MAN data, preventing a true global analysis, through satellite product intercomparisons.



All satellite daily AOD show good to very good correlations with AERONET ($0.73 \leq r \leq 0.89$), while global biases vary between -0.04 and 0.04. In 3-year averaged AOD, site specific biases can be as high as 50% (either positive or negative), although a more typical value is 15% for the top performing products and 25% for the less performing products (in absolute values: 0.025 to 0.040). These site specific biases show regional patterns of varying sign that together cause a balancing of errors in the traditional global bias estimate of satellite AOD, which may not be a very useful metric for satellite AOD performance. In addition to these biases, satellite products also exhibit random errors that appear to be at least 1.6 to 3 times larger than the site specific biases. Evaluation of satellite products on a daily time-scale (dominated by random errors rather than biases) therefore gives only limited information on the usefulness of a product for global multi-year model evaluation. While evaluation results for AERONET are usually robust, considerable uncertainty remains in the evaluation by MAN data due to the low data count (3 years of data).

The satellite intercomparison confirms the previous evaluation but extends its spatial scope. Daily satellite data usually correlates very well with other satellite products, and 3-year averages show regional patterns in product differences. These patterns can often, but not always, be linked to major orography. In any case, the patterns show large spatial scales which should aid in the identification of their causes. Over ocean, product differences are both smaller and spatially smoother, with a latitudinal dependence. The best agreement in AOD is found when using the same algorithm for the same sensor on two different platforms (Terra/Aqua). Large differences in AOD can be found for products using different algorithms but the same platform and sensor (MODIS on either Aqua or Terra, or AATSR on ENVISAT). Already variations in the same algorithm can lead to substantially different AOD (DeepBlue for MODIS, AVHRR and SeaWiFS).

Although the aggregated AOD correlates quite well among satellite products, we were able to show that the area covered in each $1^o \times 1^o$ grid-box (called: spatial coverage), correlates significantly less well among the products. We present evidence that this spatial coverage is determined mostly by (observed) cloud fraction and suggest there may be substantial differences in the quality of cloud screening by the different products. Product differences at low spatial coverage (high cloud fraction) are about twice as large than at high spatial coverage (low cloud fraction).

Intercomparing the product evaluation (with AERONET) of satellite products is challenging. A true apples-to-apples comparison requires collocating all datasets (including AERONET) but this greatly reduces the number of data available for analysis. As a consequence, it is likely that those data sample only part of the underlying true error distribution. We showed that an apples-to-apples comparison results in different results (from an individual collocation with AERONET) for some datasets, but no great changes for others. As we were able to show this is unlikely the result of statistical noise, we seem forced to conclude that a true comparison of product skill is only possible for a limited set of circumstances.

Collocating either the morning or afternoon products together allows to create maps of 3-year averaged AOD diversity amongst the products. Although there are differences, the diversity for morning and afternoon products shows similar patters and magnitudes. Diversity shows a lot of spatial variation, from $10\%$ over parts of the ocean to $100\%$ over parts of central Asia and Australia. Also, in a broad statistical sense, diversity can be shown to relate to retrieval signal-to-noise and uncertainty in cloud masking within the $1^o \times 1^o$ grid-boxes of super-observations. The most interesting find, however, is that diversity can be used to predict uncertainty in 3-year averaged AOD of individual satellite products (at least for the better performing products).



The possible applications of diversity and its interpretation as uncertainty are multiple. First, diversity shows (by definition) where satellite products differ most and thereby offer clues on how to improve them. Second, for the same reason, diversity may be used as guidance in choosing future locations for AERONET sites. Observations at locations with large diversity offer more information on individual satellite performance than those from locations with small diversity. Third, diversity as uncertainty provides a spatial context to the product evaluation with AERONET. Fourth, and related to Third, diversity as uncertainty offers a very simple way to evaluate & intercompare new satellite products to the 14 products considered in this study. To perform better than these products, their normalized 3-year difference from AERONET (Eq. 3) should exhibit a standard deviation smaller than 1 (see Fig. 30). Fifth, again related to Third, diversity as uncertainty offers modellers a simple estimate of the expected multi-year average uncertainty in satellite AOD.

*Code and data availability.* All remote sensing data is freely available. Analysis code was written in IDL and is available from the author upon request.

*Author contributions.* NS designed the experiments, with the help of GL, TP, SK, MS and PS, and carried them out. AS, AH, CH, HJ, PL, RL, AL, AL, PN, CP, VS, LS, GT, OT, and YW provided the data. NS prepared the manuscript with contributions from all co-authors

*Competing interests.* No competing interests are present

*Acknowledgements.* We thank the PI(s) and Co-I(s) and their staff for establishing and maintaining the many AERONET sites used in this investigation. The figures in this paper were prepared using David W. Fanning's Coyote Library for IDL. The work by NS is part of the Vici research programme with project number 016.160.324, which is (partly) financed by the Dutch Research Council (NWO). PS acknowledges funding from the European Research Council (ERC) project constRaining the EffeCts of Aerosols on Precipitation (RECAP) under the European Union's Horizon 2020 research and innovation programme with grant agreement No 724602, the Alexander von Humboldt Foundation and from the Natural Environment Research Council project NE/P013406/1 (A-CURE).





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





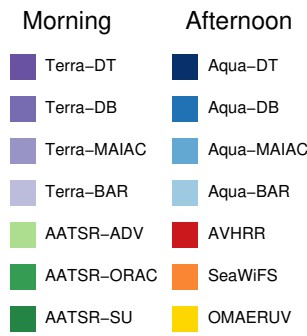

**Figure 1.** Colour legend used throughout this paper to designate the different satellite products, organised by approximate local equator crossing time.

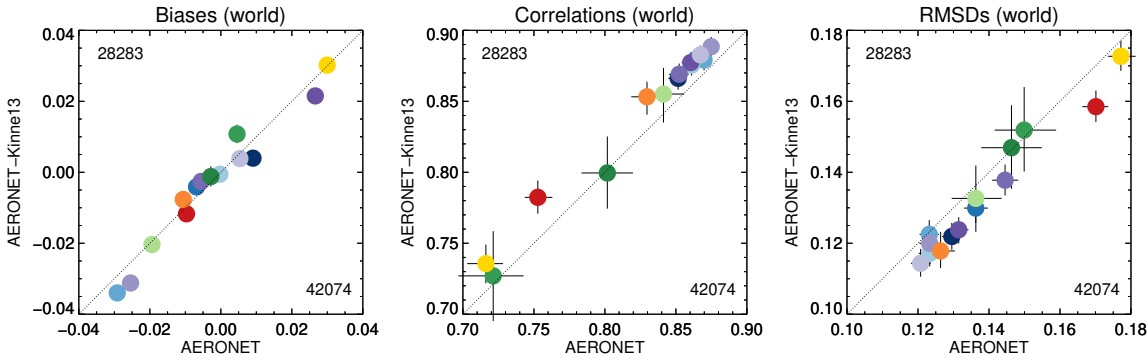

**Figure 2.** Comparison of the evaluation of satellite products for different selections of AERONET sites. Horizontally: results using all AERONET sites; vertically: using the Kinne et al. 2013 selection. Colours indicate satellite product, see also Fig 1. Numbers in upper left and lower right corner indicate amount of collocated data, averaged over all products. Collocation of individual datasets with AERONET within 1 hour. Error bars indicate 5-95% uncertainty range based on a bootstrap analysis of sample size 1000.





**Table 1.** Papers intercomparing multiple satellite datasets

| Reference | Period | Resolution Temporal | Spatial | Region | AERONET | Sensors |
|---|---|---|---|---|---|---|
| Myhre et al. (2004) | Nov 1996 - Jun 1997 | monthly | 1° | ocean | 13 | AVHRR-1,-2, OCTS, POLDER, TOMS |
| Myhre et al. (2005) | Sept 1997 - Dec 2000 | monthly | 1° | ocean | 33 | ATSR, AVHRR-1,-2, MODIS (2×), MISR, SeaWiFS, TOMS, VIRS |
| Kinne (2009) | 1981-2005 (partially overlapping) | monthly | ? | globally | 264 | AVHRR, MISR, MODIS (2×), POLDER, TOMS |
| Bréon et al. (2011) | 2004 - 2011 | 30$^{min}$ | 50 km | globally | ∼ 200 | MERIS, MODIS (2×), POLDER, SEVIRI |
| Holzer-Popp et al. (2013) | Sept 2008 | daily | 1° | globally | unknown | AATSR (3×), MERIS, PARASOL |
| Petrenko and Ichoku (2013) | 2006-2010 | 30$^{min}$ | 55 km | globally | 393 | MISR, MODIS (2×), OMI, POLDER, SeaWiFS |
| Leeuw et al. (2015) | 4 months in 2008 | daily | 1° | globally | unknown | AATSR (3×), MERIS, MODIS-TERRA, POLDER |
| Sogacheva et al. (2019) | 15 years (partially overlapping) | monthly | 1° | globally | unknown | AATSR (3×), ATSR-2 (3×), AVHRR, EPIC, GRASP, MISR, MODIS (4×), OMAERUV, SeaWiFS, TOMS, VIIRS |

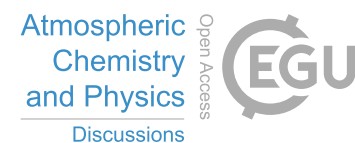

**Table 2.** Remote sensing products used in this study

| Platform | Overpass [hr] | Sensor | Swath [km] | Pixel [km] | Product | AOD[1] 550nm | Comments | References |
|---|---|---|---|---|---|---|---|---|
| Terra | 10:30AM | MODIS | 2330 | 1 | Dark Target C6.1 | R | Terra-DT | Remer et al. (2005) |
| | | | | | Deep Blue C6.1 | I/E | Terra-DB | Hsu et al. (2013, 2019); Sayer et al. (2019b) |
| | | | | | MAIAC v2.0 | E | Terra-MAIAC | Lyapustin et al. (2018) |
| | | | | | BAR v1.0 | R | Terra-BAR | Lipponen et al. (2018) |
| ENVISAT | 10:30AM | AATSR | 500 | 1 | ADV/ASV Ver2.30 | R | AATSR-ADV | Sogacheva et al. (2017) |
| | | | | | ORAC v03.20 | R | AATSR-ORAC | Thomas et al. (2009) |
| | | | | | AARDVARC v4.21 | R | AATSR-SU | North et al. (1999); North (2002); Bevan et al. (2012) |
| Aqua | 1:30PM | MODIS | 2330 | 1 | Dark Target C6.1 | E | Aqua-DT | see Terra-DT |
| | | | | | Deep Blue C6.1 | I/E | Aqua-DB | see Terra-DB |
| | | | | | MAIAC v2.0 | E | Aqua-MAIAC | see Terra-MAIAC |
| | | | | | BAR v1.0 | R | Aqua-BAR | see Terra-BAR |
| SeaSTAR | 0:20PM | SeaWiFS | 1502 | 13.5 | Deep Blue & SOARv004 | I/E | SeaWiFS | Hsu et al. (2013); Sayer et al. (2012a, b) |
| noaa18 | 2:58PM | AVHRR | 2900 | 8.8 | Deep Blue & SOAR v001 | E R | AVHRR | Hsu et al. (2017); Sayer et al. (2017) |
| AURA | 1:30PM | OMI | 2600 | 18 | OMAERUV v1.7.1 | E | OMAERUV | Ahn et al. (2014); Jethva et al. (2014) |

1) Interpolated or Extrapolated to 550 nm, depending on surface type; or Retrieved at 550 nm



**Table 3.** AERONET site subsets

| Reference | Criterion | Nr of sites |
|---|---|---|
| All sites | mountain sites included | 1144 |
| Kinne et al. (2013) | sites with high maintenance ($q \geq 2$), mountain sites removed | 255 |
| Schutgens (2019) | sites with yearly representation error $\leq$ 20%, mountain sites above 1500 m removed | 859 |

Kinne et al. (2013) considers only sites before 2009, with at least 5 months of data.

**Table 4.** Averaged product evaluation with AERONET depending on selection AERONET sites used as truth reference.

| metric | all | Kinne et al. (2013) | Kinne et al. (2013) (pruned) | Schutgens (2019) |
|---|---|---|---|---|
| bias | -0.0024 | -0.0031 | -0.0031 | -0.001 |
| correlation | 0.826 | 0.841 | 0.841 | 0.845 |
| RMSD | 0.139 | 0.133 | 0.134 | 0.136 |
| nr of obs | 42074 | 28283 | 28150 | 32716 |

**Table 5.** Averaged product evaluation with AERONET depending on temporal constraints (pruned Kinne subset)

| metric | $\Delta$t=1, n=2 | $\Delta$ t=3, n=1 | $\Delta$t=3, n=3 | $\Delta$t=3, n=5 | Sect. 7 |
|---|---|---|---|---|---|
| bias | -0.0030 | -0.0021 | -0.0026 | -0.0031 | -0.017 |
| correlation | 0.850 | 0.833 | 0.847 | 0.858 | 0.823 |
| RMSD | 0.125 | 0.138 | 0.130 | 0.120 | 0.100 |
| nr of obs | 21938 | 31129 | 25558 | 18412 | 3986 |





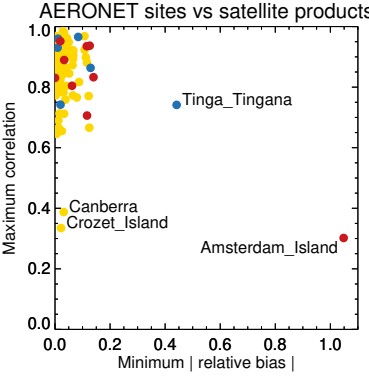

**Figure 3.** Minimum relative bias (sign-less) and maximum correlation per AERONET site, over all products. Red symbols indicate AERONET site bias is always positive, blue symbols indicate AERONET site bias is always negative. Yellow symbols indicate that site bias is positive versus some products, and negative versus others. Products were individually collocated with AERONET (Kinne et al. 2013 selection) within 1 hour.

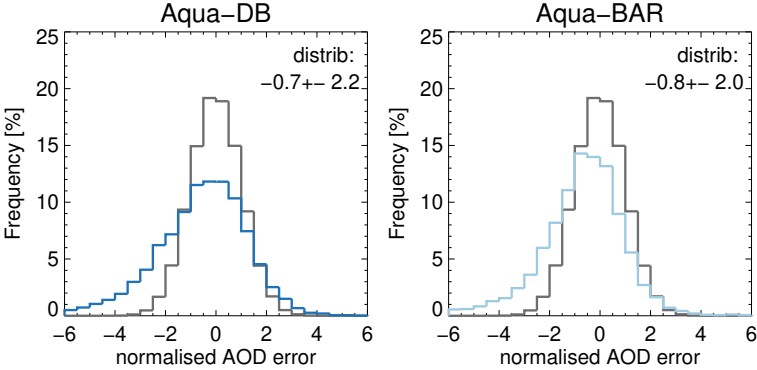

**Figure 4.** The normalized satellite AOD error, as defined in Eq. 2 for the Aqua-DeepBlue and BAR products (in blue), for cases where the spatial coverage $\geq 80\%$. A Gaussian distribution with zero mean and standard deviation of 1 is shown in black. The normalized error appears to be significantly larger than the squared sum of the representation error and AERONET observation error, suggesting that satellite observation errors dominate. The values in the top-right corner are mean and standard deviation of the normalized error. Products were individually collocated with AERONET (Kinne et al. 2013 selection) within 1 hour.





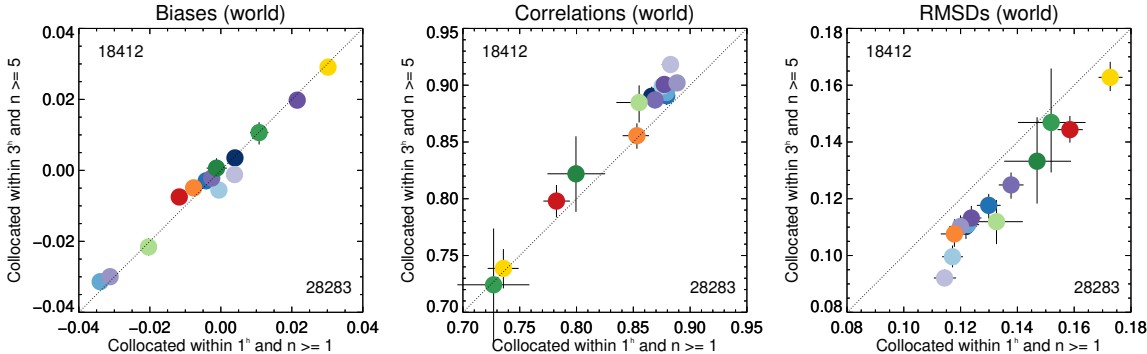

**Figure 5.** Comparison of the evaluation of satellite products depending on collocation criterium. Horizontally: results using at least one AERONET observation within 1 hour; vertically: using at least 5 AERONET observations within 3 hours. Colours indicate satellite product, see also Fig 1. Numbers in upper left and lower right corner indicate amount of collocated data, averaged over all products. Individual collocation of datasets with AERONET (Kinne et al. 2013 subset, pruned) within 1 hour. Error bars indicate 5-95% uncertainty range based on a bootstrap analysis of sample size 1000.

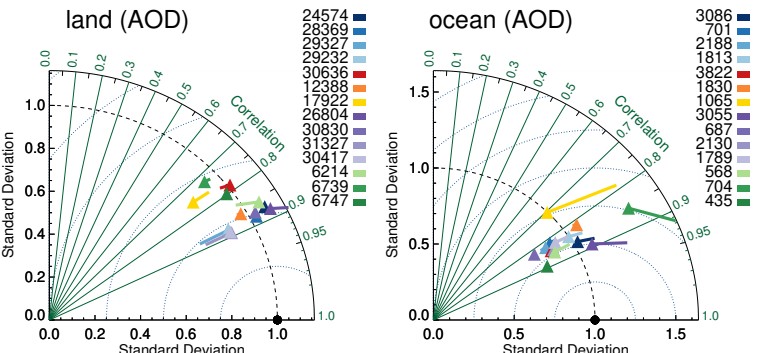

**Figure 6.** Taylor diagram for satellite products evaluated over either land or ocean with AERONET. Symbols indicate correlation and internal variability relative to AERONET, the line extending from the symbol indicates the (normalized) bias (see also Sect. 3.1). Colours indicate satellite product (see also Fig. 1), numbers next to coloured blocks indicate amount of collocated data. Products were individually collocated with AERONET (Kinne et al. 2013 selection, pruned) within 1 hour.





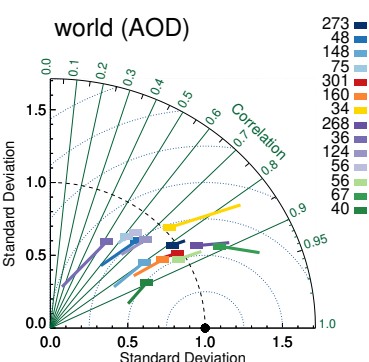

**Figure 7.** Taylor diagram for satellite products evaluated over either land or ocean with MAN. Symbols indicate correlation and internal variability relative to MAN, the line extending from the symbol indicates the (normalized) bias (see also Sect. 3.1). Colours indicate satellite product (see also Fig. 1), numbers next to coloured blocks indicate amount of collocated data. Products were individually collocated with AERONET (Kinne et al. 2013 selection, pruned) within 1 hour.

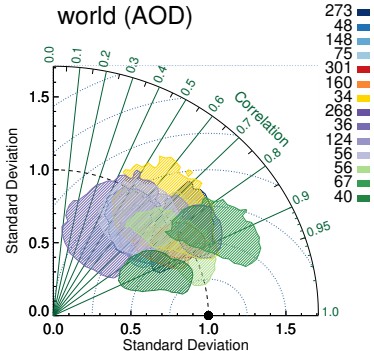

**Figure 8.** Taylor diagram for satellite products evaluated with MAN. Same as Fig. 7 except regions indicate $5 - 95\%$ uncertainty range in correlation and standard deviation from a bootstrap analysis of sample size 10000.





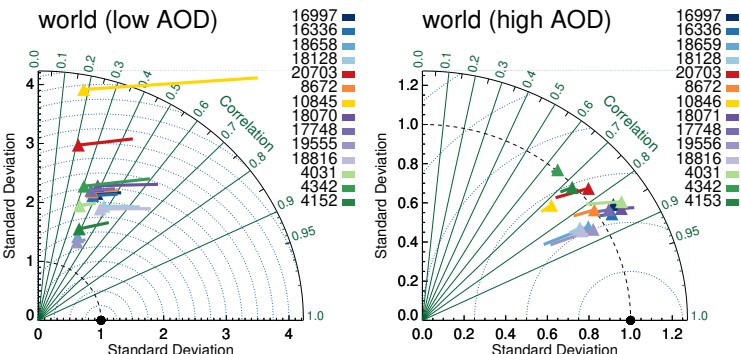

**Figure 9.** Taylor diagram for satellite products evaluated with AERONET at either low or high AOD (distinguished by median AERONET AOD ∼ 0.12). Symbols indicate correlation and internal variability relative to AERONET, the line extending from the symbol indicates the (normalized) bias (see also Sect. 3.1). Colours indicate satellite product (see also Fig. 1), numbers next to coloured blocks indicate amount of collocated data. Products were individually collocated with AERONET (Kinne et al. 2013 selection, pruned) within 1 hour.

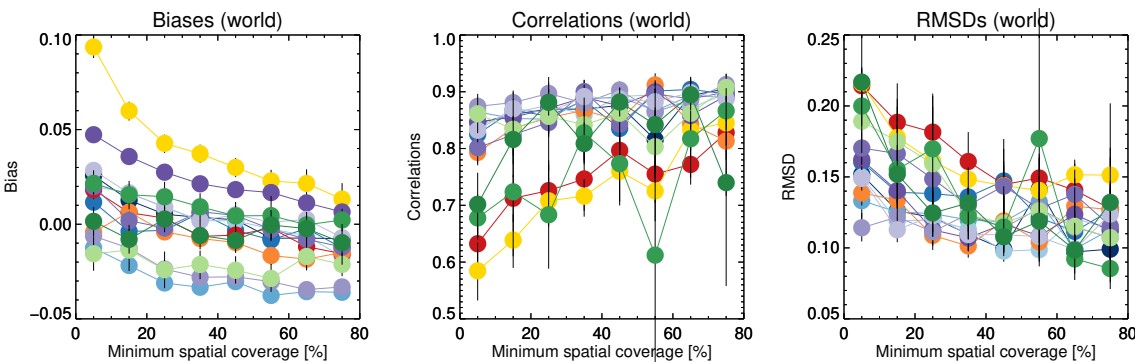

**Figure 10.** Evaluation of satellite products with AERONET, binned by minimum spatial coverage. Colours indicate satellite product, see also Fig 1. Individual collocation of datasets with AERONET (Kinne et al. 2013 selection, pruned) within 1 hour. Error bars indicate 5-95% uncertainty range based on a bootstrap analysis of sample size 1000.



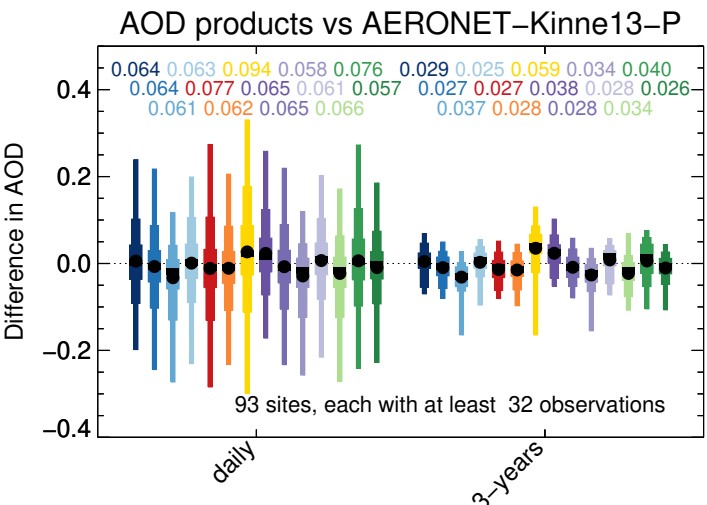

**Figure 11.** Evaluation of satellite products with AERONET, either for daily data or 3-year averages. Box-whisker plot shows 2, 9, 25, 75, 91 and 98% quantiles, as well as median (block) and mean (circle). Numbers above the box whiskers indicate mean sign-less product errors. Colours indicate satellite product, see also Fig. 1. Products were individually collocated with AERONET (Kinne et al. 2013 selection, pruned) within 1 hour. All products use the same sites, each of which produced at least 32 collocations with each product.

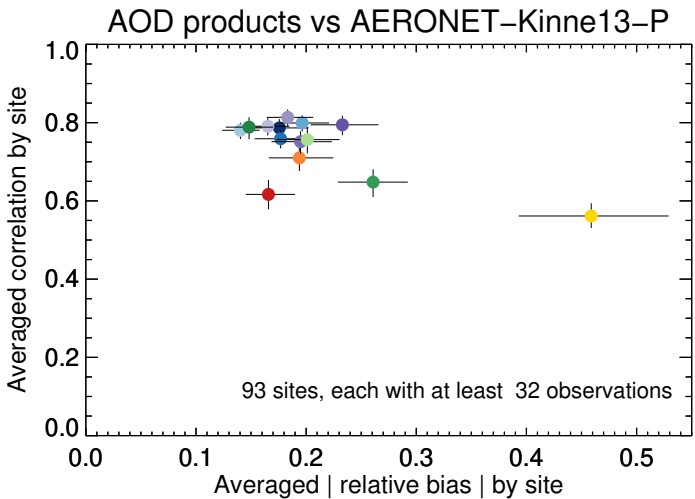

**Figure 12.** Evaluation of satellite products with AERONET per site, averaged over all sites. Error bars indicate 5-95% uncertainty range based on a bootstrap analysis of sample size 1000. Colours indicate satellite product, see also Fig. 1. Products were individually collocated with AERONET (Kinne et al. 2013 selection, pruned) within 1 hour. All products use the same sites, each of which produced at least 32 collocations with each product.





**Figure 13.** For MODIS-Aqua products are shown: a scatter plot of individual super-observations versus AERONET (mean and standard deviation refer to the difference with AERONET, PCorr and OLSB refer to the linear correlation and a robust least squares estimator of the regression slope); the AOD difference for individual super-observations as a function of spatial coverage (individual data, sub-sampled to a 1000 points, are shown as black dots using the left-hand axis, while the distribution per coverage bin, in grey-scales indicating 2, 9, 25, 75, 91, and 98% quantiles, uses the right-hand axis); a global map of the three-year AOD average; a global map of the three-year AOD difference average with AERONET (if site provided at least 32 observations; land sites are circles, ocean sites are squares, diamonds are the remainder). Products were individually collocated with AERONET (Kinne et al. 2013 selection, pruned) within 1 hour.



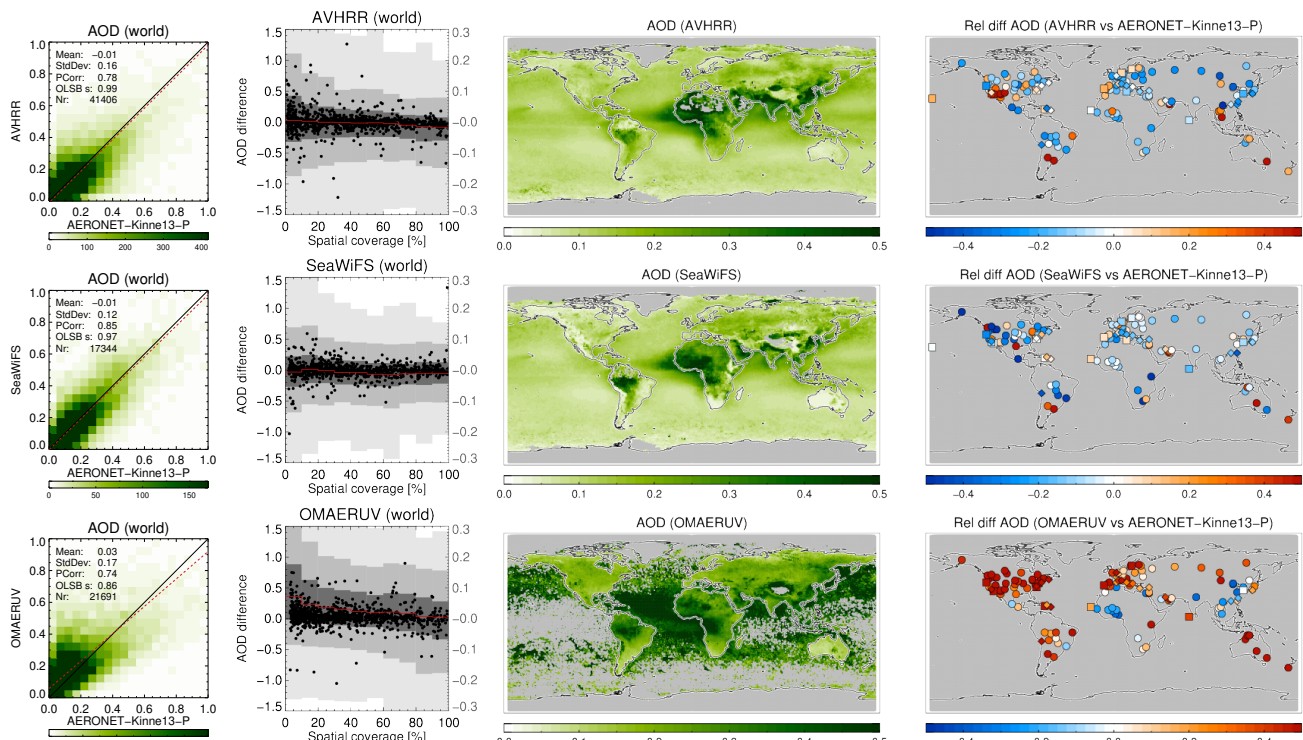

**Figure 14.** Same as Fig. 13, for AVHRR, SeaWiFS and OMAERUV products.





**Figure 15.** Same as Fig. 13, for MODIS-Terra products.





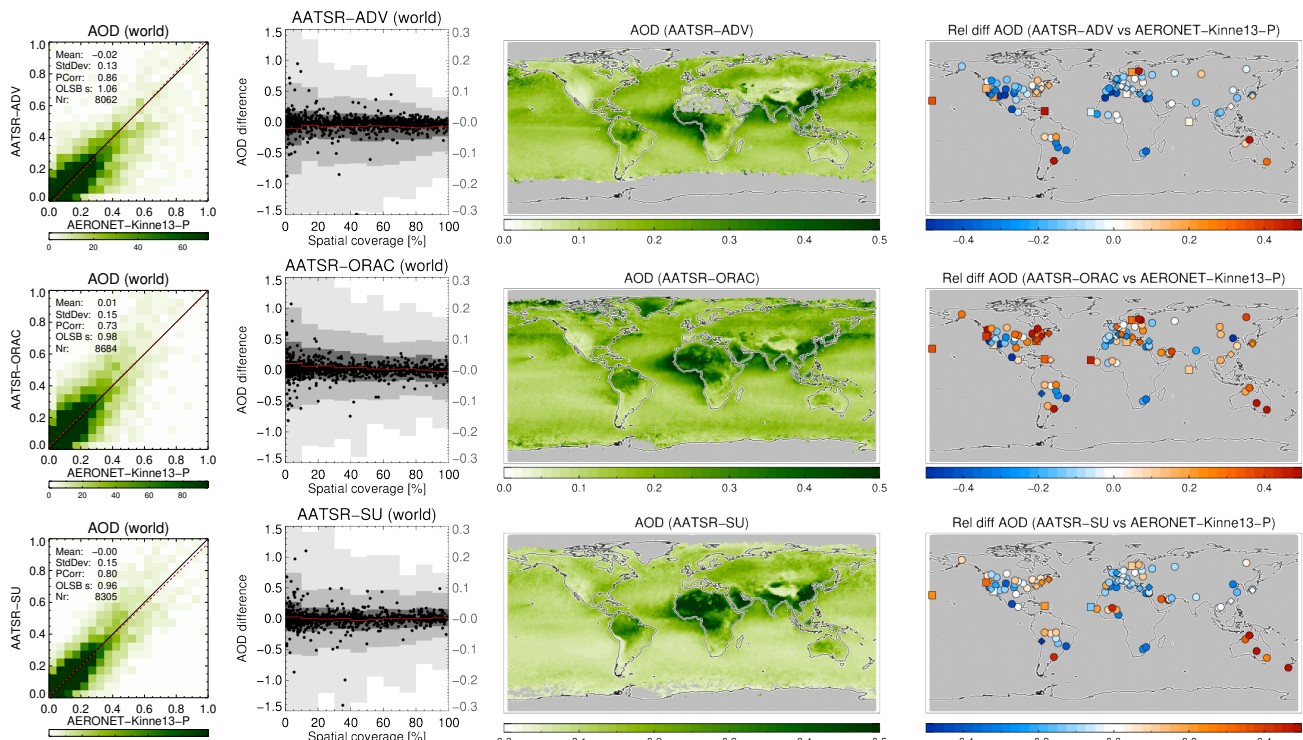

**Figure 16.** Same as Fig. 13, for AATSR products.





**Figure 17.** Global maps of the 3-year averaged difference in AOD for satellite products on morning satellites. Products were pair-wise collocated within 1 hour.





**Figure 18.** Global maps of the 3-year averaged difference in AOD for satellite products on afternoon satellites. Products were pair-wise collocated within 1 hour.





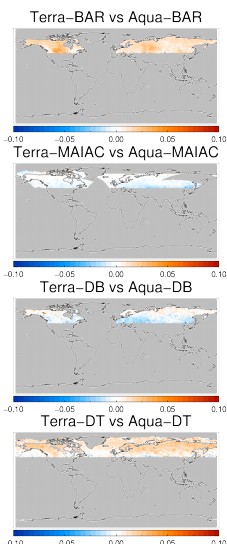

**Figure 19.** Global maps of the 3-year averaged difference in AOD for products based on the same algorithm and either Aqua and Terra satellites. Products were pair-wise collocated within 1 hour.

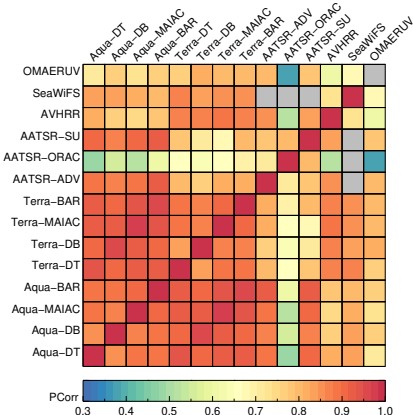

**Figure 20.** Correlation of AOD super-observations for satellite products. Products were pair-wise collocated within 1 hour.



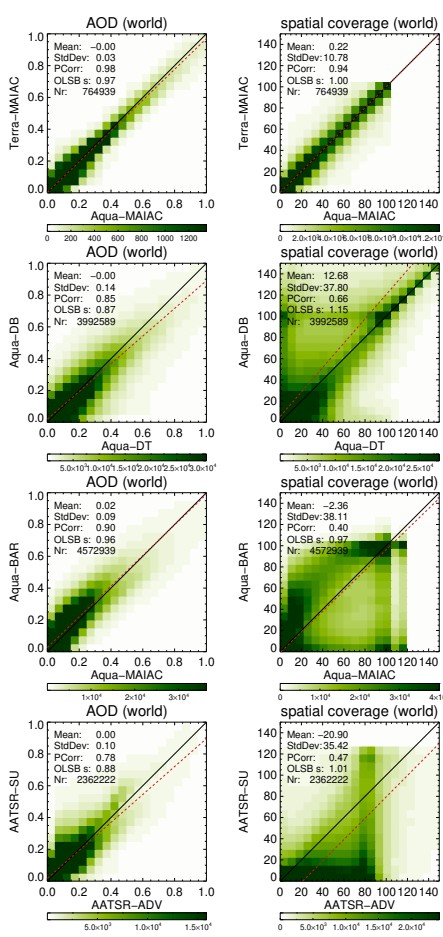

**Figure 21.** Scatterplot of AOD and spatial coverage from super-observations for selected satellite products. Products were pair-wise collocated within 1 hour.





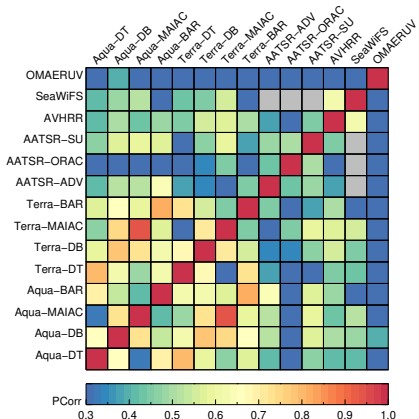

**Figure 22.** Correlation of spatial coverage in super-observations for satellite products. Products were pair-wise collocated within 1 hour.

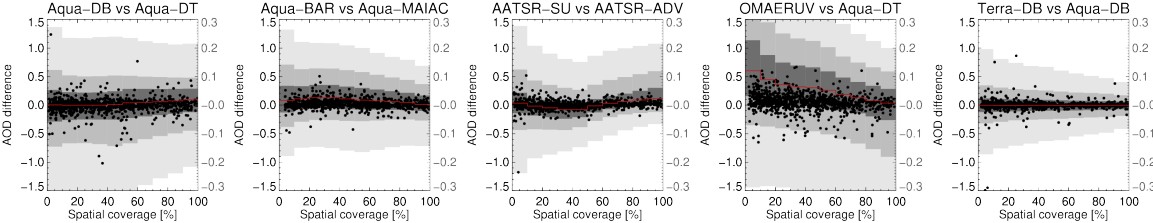

**Figure 23.** AOD difference between super-observations for selected products as a function of spatial coverage (here the average of the two products). Individual data are shown as black dots (using left axis) while distributions per coverage bin are shown as grey scales (2, 9, 25, 75, 91 and 98% quantiles, using right-hand axis). Products were pair-wise collocated within 1 hour.



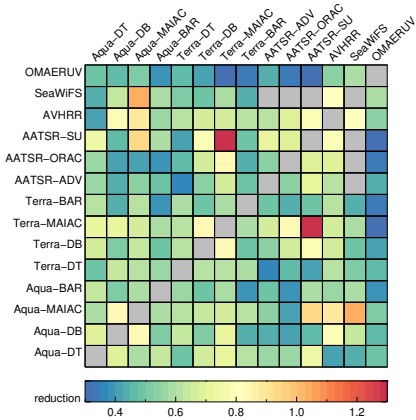

**Figure 24.** The ratio of typical difference (mean of sign-less difference) for spatial coverage at $90-100\%$ to $0-10\%$. Products were pair-wise collocated within 1 hour.

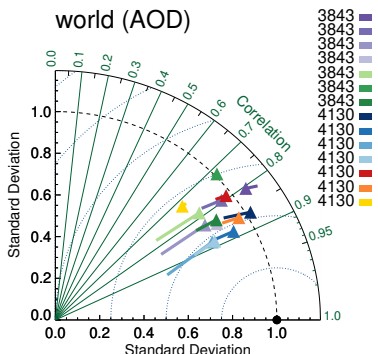

**Figure 25.** Taylor diagram for satellite products evaluated with AERONET. Symbols indicate correlation and internal variability relative to AERONET, the line extending from the symbol indicates the (normalized) bias (see also Sect. 3.1). Colours indicate satellite product (see also Fig. 1), numbers next to coloured blocks indicate amount of collocated data. All morning products were collocated together with AERONET (Kinne et al. 2013 selection, pruned) within 3 hours, similar for all afternoon products.





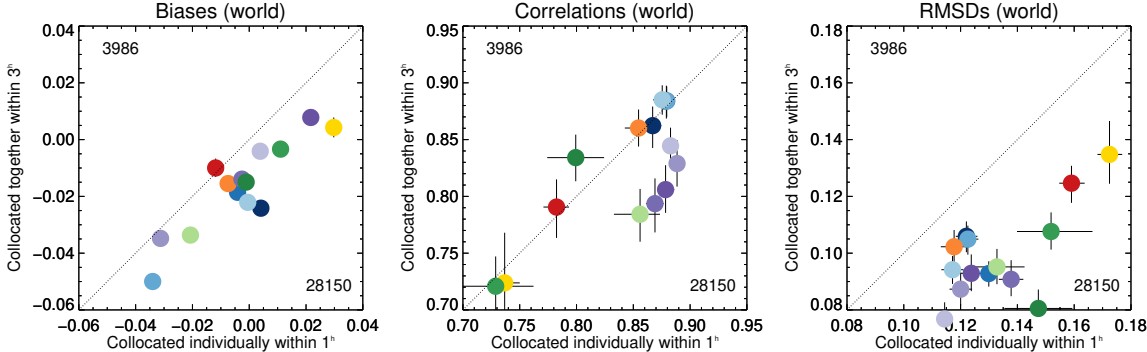

**Figure 26.** Comparison of the evaluation of satellite products when collocating individual products with AERONET (within 1 hour) or all morning or afternoon products with AERONET (within 3 hours). Colours indicate satellite product, see also Fig 1. Numbers in upper left and lower right corner indicate amount of collocated data, averaged over all products. The AERONET data are the Kinne et al. 2013 selection, pruned. Error bars indicate 5-95% uncertainty range, based on a bootstrap sample of 1000.

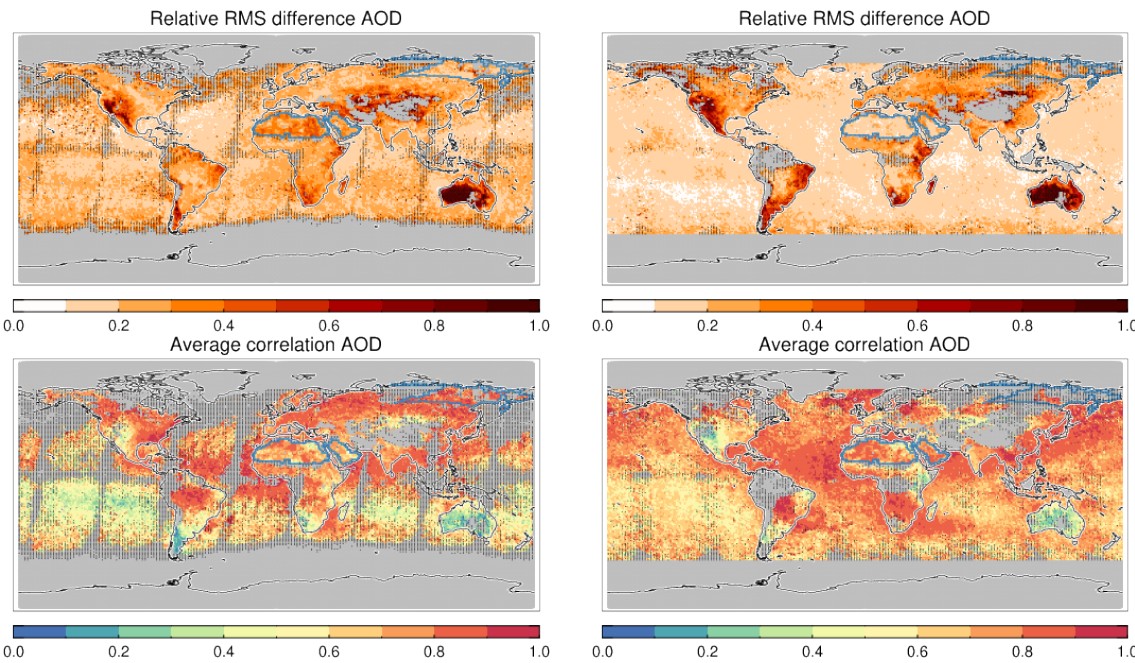

**Figure 27.** Global maps of relative diversity and average correlation of collocated satellite products. Diversity is the spread in AOD over the mean AOD. The average correlation is the average over all pair-wise correlations possible. Dotted areas indicate that the uncertainty due to statistical noise (standard deviation) is at least 0.1 (or less than 10 super-observations for each product were available). Morning (left) and afternoon (right) products were collocated within 3 hours.



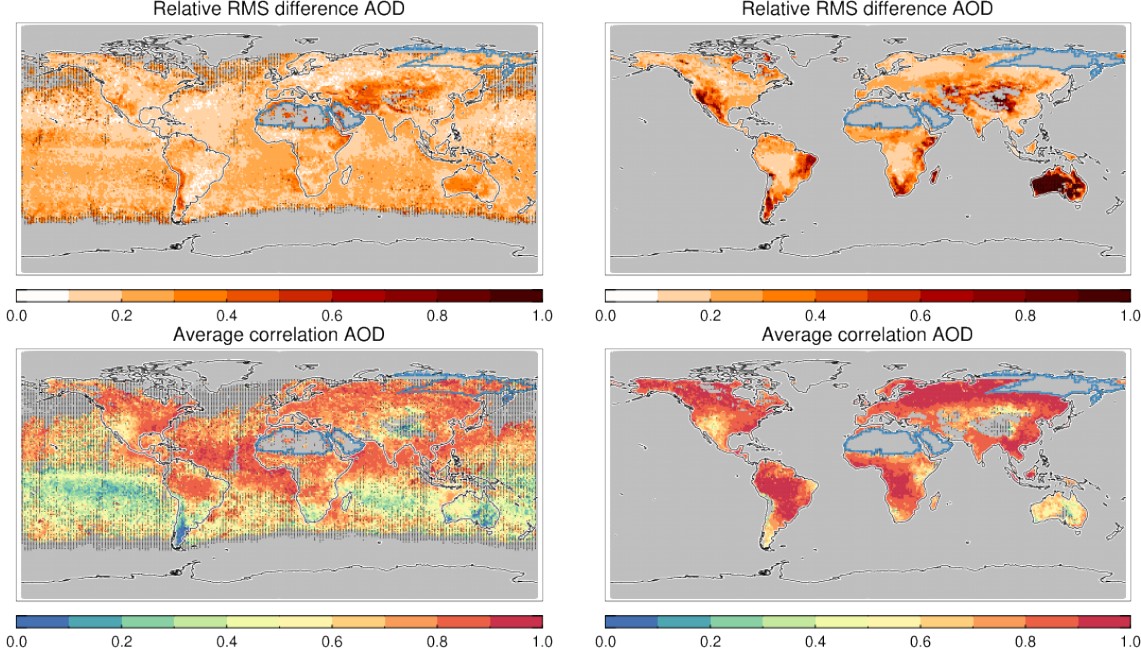

**Figure 28.** Same as Figure 27 but different selections of satellite products . Morning AATSR (left) and afternoon Aqua (right) products were collocated within 3 hours.

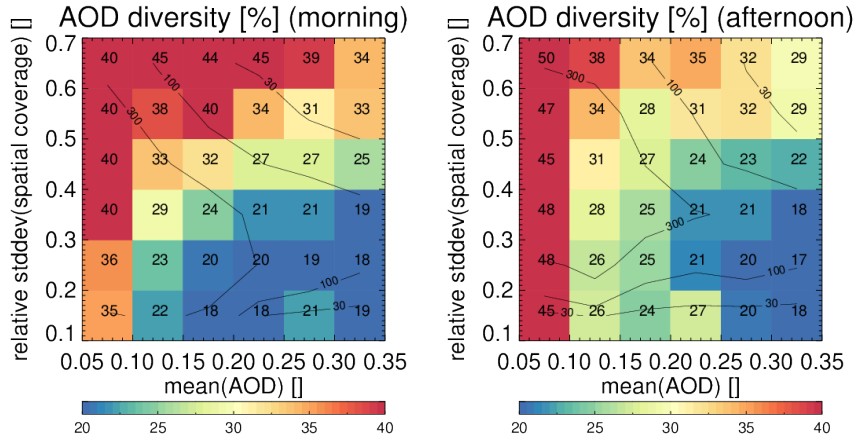

**Figure 29.** Diversity in AOD amongst morning and afternoon products as a function of mean AOD and the relative standard deviation in spatial coverage. The values in each bin show averaged diversity (similar to the colour). The contour lines show data density. Morning (right) and afternoon (left) products were collocated within 3 hours. The statistics are dominated by observations over land. Over ocean, similar patterns are found but the range in diversity is much reduced.





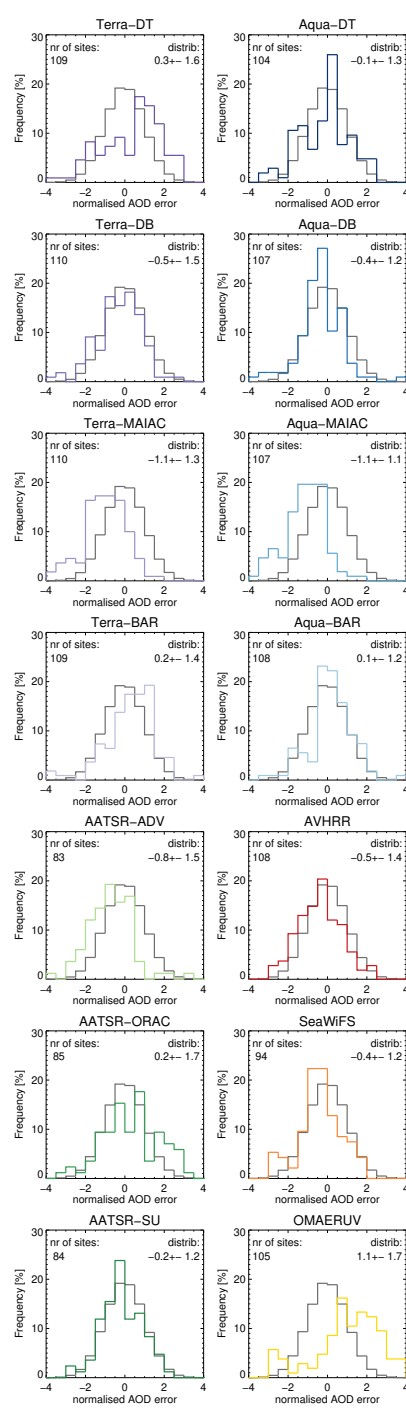

**Figure 30.** The 3-year averaged AOD error distributions, normalized to the diversity (spread in the ensemble, see Fig. 27). Errors are based on individual collocations of products with AERONET, unlike the diversity which is based on collocation of either all morning or afternoon satellite products together. Mean and standard deviation of the product's distribution are shown in the upper left corner. Only sites with at least 32 observations were used. For comparison, a normal distribution with mean zero and standard deviation 1 is also shown (in black).