# Peer review of "An AeroCom/AeroSat study: Intercomparison of Satellite AOD Datasets for Aerosol Model Evaluation"

_Atmospheric Chemistry and Physics, 2019_

## Short Comment (SC1) · 15 Feb 2020

I think this is an interesting paper. This paper did not mention a recent study by Wei et al., (2019) who did a comparison of 11 global monthly AOD products. I could not see it in the Table 1 of the current study? I think it would be useful to provide some comparison of the current study with Wei et al., 2019.

Wei, J., Peng, Y., Mahmood, R., Sun, L., and Guo, J.: Intercomparison in spatial distributions and temporal trends derived from multi-source satellite aerosol products, Atmos. Chem. Phys., 19, 7183–7207, https://doi.org/10.5194/acp-19-7183-2019, 2019.

---

## Referee Comment (RC1) · Knut Stamnes (Referee) · 20 May 2020

Review of manuscript submitted to ACP:

Title: An AeroCom/AeroSat study: Intercomparison of Satellite AOD Datasets for Aerosol Model Evaluation

Authors: Nick Schutgens et al.

**General Comments**

In this paper, the authors evaluate and compare fourteen satellite (seven morning and seven afternoon) products, representing 9 different retrieval algorithm families using observations from 5 different sensors on 6 different platforms. The five sensors are MODIS (Terra and Aqua), AATSR, SeaWiFS, AVHRR, and OMI. The evaluation is done for three years (2006, 2008, and 2010) by using AERONET (AErosol RObotic NETwork) and MAN (Maritime Aerosol Network) data as "ground truth". The results show that aggregated satellite AOD agrees much better than the spatial coverage (often driven by cloud masks) and that up to 50% of the difference between satellite AOD is attributed to cloud contamination.

Relative diversity, defined as the standard deviation divided by mean, is calculated at each grid-box from the 3-year average AOD values of all seven morning or afternoon products. The point is being made that this diversity may be used as an indication of AOD uncertainty.

The paper is well organized, well written, and the results are presented in a clear and concise manner. Nevertheless, the paper could be strengthened in the following ways:

1. In view of the fact that cloud masking seems to be one of the primary reasons for the discrepancies between AOD results, a discussion of how the various algorithms deal with cloud screening would be of great interest.

2. Another important reason for the reported discrepancies is attributed to the treatment of reflection by the underlying surface. This issue deserves some attention in the paper.

3. There is no discussion of the various algorithms, which is a weakness because the average reader that the authors may want to reach will not be familiar with these algorithms. Therefore, a brief description of each of the algorithms is highly recommended to be included in Section 2 of the paper. This addition would make the paper of interest to a much wider audience, and it should include a description of how each algorithm deals with cloud screening and surface reflection.

**Specific Comments**

- On page 4 the authors state: "We will provide evidence that cloud masking is the dominant factor....". Please be more specific about where in the paper such evidence is provided.

- On page 5 the authors state: "Most ocean boxes with observations will be in coastal regions, with some over isolated islands." Please explain the reason for this restriction.

The MAN web-site indicates that data are obtained by Mircrotops deployed on ships all over the world. Why are those data not used in this study?

- On page 6, the following sentence appears: "Bootstrapping has been shown to be reliable even for relatively small sample sizes." A reference in support of this statement seems to be required.

- on page 9 the authors remark: "It is not surprising that some products will have better cloud masking than others ...". Some explanation would be useful here.

- On page 10 the authors write: "Terra/Aqua-DT, by the way, sometimes produces negative AOD leading to e.g. very low values for averaged AOD over Australia." What is the reason for this problem?

- On bottom of page 10: "The contrasts in the differences over land and neighbouring ocean (e.g. African outflow for Terra-DT with AATSR-SU or AATSR-ADV, or Aqua-DT with OMAERUV, or AVHRR with SeaWiFS) may likewise be driven by albedo treatment." Some more discussion of this "albedo treatment" issue would be useful.

- On page 11 the authors write: "The three products based on the DeepBlue algorithm (Aqua-DB, AVHRR and SeaWiFS) suggest that already small algorithmic differences can yield significant differences." Please be more specific about what is meant by "small algorithmic differences".

- On page 12, the authors state: "Diversity is generally lowest over ocean, never reaching over 30% while over land values of 100% are possible." Please explain this result.

- On page 13, the authors state: "We see that the diversity goes down when the mean AOD increases, and goes up when the uncertainty in cloud masking increases. This is as one would expect." Please elaborate on why this result is to be expected.

**Technical Corrections**
In general this paper is well written, and I did not spot any typographical or grammar mistakes, except for the following:
at the top of page 3, the following text appears:
...the data usually come in different spatio-temporal grids. In addition, data has often....
The noun "data" is plural (from the singular latin word "datum"). In the first appearance "data usually come" it is considered to be plural, while in the second appearance "data has" it is considered to be singular. I would recommend that "data" be considered to be plural, and that the whole paper be searched and changed to amend this inconsistency.

---

## Short Comment (SC2) · 25 May 2020

Dear Rashed,

Thanks for pointing out this paper (which I somehow missed), I will include in the final version of our manuscript.

Nick

---

## Referee Comment (RC2) · Anonymous Referee #2 · 19 Jun 2020

This manuscript provides an evaluation and intercomparison of 14 different satellite AOD products, based on 9 different retrieval algorithm families using observations from 5 different sensors on 6 different platforms. The 14 satellite products include retrievals from MODIS (Terra/Aqua), AATSR (ENVISATE), AVHRR (NOAA18), SeaW-iFS (SeaStar) and OMI (Aura). The validation is made with AERONET and MAN data for 2006, 2008 and 2010 three years on daily and multi-year time scales. With the increased numbers of available satellite AOD products in the past two decades, there is a need for users to know uncertainties of these products and how they intercompare using a same evaluation method. This research provides such very needed informa-tion. Besides basic verification statistics (bias, correlation, RMSE), the authors also

discuss spatial and temporal sampling impacts on verification results. One of their findings is that the diversity among these products may be used as an indication of AOD uncertainty for the better performing satellite products. This would be potentially useful for satellite AOD applications, e.g, AeroCOM model verifications over areas without AERONET measurement. Overall the manuscript is well organized and statistical analysis is nicely carried out.

I had a chance to read the other review comment, and I am in general agreement with the comments there. I strongly agree that there could be brief descriptions for each individual AOD algorithms, and how each one of them treats clouds and surface. This information could be put in an appendix. As "Up to 50% of the difference between satellite AOD is attributed to cloud contamination", knowing how each individual algorithm treats clouds is very important to help understand the AOD differences. On page 9 line 8-9 "In contrast, it is hard to use other factors determine spatial coverage (sun-glint, surface albedo, failed retrievals) to explain this."("this" here means impact of spatial coverage upon evaluation result) The authors look only at global scale. However it is expected that the importance of surface albedo may show up in some regions, e.g., mountainous regions. So it may be worth some regional analysis on the impact of spatial coverage upon evaluation result. In addition, the detailed analysis is acknowledged, however a total number of 30 for figures is relatively high. Authors could consider moving some figures into supplement, and making the most important results stand out in the manuscript.

Some minor points are listed below:

It is noted that the author has his own writing style, which is fluent, however, not necessarily formal. For example, the second sentence on Page 13 line 15, starts with "E.g." which should be "For example. . .." And there are many more places which are not listed in this review. I would leave to the editor if minor English editing is required.

P1 Line 10: It is confusing what "spatial coverage" means here. Please be specific.

[Figure]

P2 Line 25: "AOD (Aerosol Optical Depth)" should be "Aerosol Optical Depth (AOD)", ie., full expression first, and abbreviation next. Same thing for Line 28, MODIS, MISR abbreviations, and AERONET.

P3 Line 8, please define "super-observation".

P4 line 27-28 this sentence reads awkward.

P6 Line 18-19, I don't think this averaging over all sites of their bias and correlation is a novel error metric.

P23 Table 1, Under "Spatial" resolution column , there is a "?" for Kinne (2009), which needs to be fulfilled.

Table 2, It would be nice to provide information about time span of each product.

Figure 5. There seem to be missing panels based on the figure caption. The figure only shows evaluation result with collocated AERONET observation within 3hours, but result with AERONET observation within 1hour is also expected.

Figure 8. Colors representing different satellite products overlap each other. For about half of the satellite products, it is impossible to see their presence. Please think of different plotting method (e.g., making hatching less dense, with different patterns, smaller area on top of larger area) so that large area does not totally cover smaller areas, etc,) to make all the products visually identifiable.

P10 Line 10 To be consistent with the rest of the manuscript, remove "FMI" in "AATSR-FMI-ADV".

Figure 24. This figure gives the ratio of difference between satellite AOD products for spatial coverage at 90-100% to 0-10%, which corresponds to approximately 0-10% to 90-100% cloud coverage if cloud is considered the largest impactor for the AOD spatial coverage. It would be nice to break up into a few similar panels, e.g, similar subplots with relatively low, median and high spatial coverages, e.g. around 10%, 30% , 50%,

70%, 90%. This information would be useful for AOD data assimilation users, as cloud fraction is one of the used information (as threshold) of AOD data to generate DA-quality product for aerosol DA. This would give some guidance on what could fraction is reasonable to obtain AOD consistency among multiple satellite products in AOD DA efforts.

Figure 27. What do the contours over north Africa, Arabian Peninsula and Siberia represent? This is explained in the text, but it would be nice to describe in the figure caption also.

Page 13 Line 8, "de average..." typo?

Figure 30 caption, typo "diveisity"

Page 14, Line 19. Summary section, "......MISR because the product was in the middle of an update cycle, and VIRRS because it was only launched in 2011." I understand the meaning of this sentence, but formal English is preferred as this is for publication. Also I believe there is a typo for VIIRS.

Page 14, line 20. "For MODIS and AATSR, four resp. three different retrieval algorithms were used". See comment above.

Page15, Line 31, "patters", typo.

---

## Author Comment (AC1) · 31 Aug 2020

Reply to reviewer 1

We would like to thank reviewer 1 (Knut Stamnes) for his time to review our paper and provide useful comments to improve the text.

General comments

The paper is well organized, well written, and the results are presented in a clear and concise manner. Nevertheless, the paper could be strengthened in the following ways:

1. In view of the fact that cloud masking seems to be one of the primary reasons for the discrepancies between AOD results, a discussion of how the various algorithms deal with cloud screening would be of great interest.

2. Another important reason for the reported discrepancies is attributed to the treatment of reflection by the underlying surface. This issue deserves some attention in the paper.

3. There is no discussion of the various algorithms, which is a weakness because the average reader that the authors may want to reach will not be familiar with these algorithms.

The reviewer asks for more details on the algorithmic procedures used in deriving fourteen different products (11, if we count Aqua-Terra as single products). While this request is understandable, we choose not to expand much further on this information for several reasons:

Practical: it would significantly increase the size of the paper which is already large. Note that we provide references to the papers that describe the individual products and their algorithms.

Scientific: such a description might make sense if we would consequently be able to use it to interpret results. However, that is not the purpose of this paper. We expect it would not be easy to make such an interpretation (see e.g. Holzer-Popp et al. 2013). We believe that the purpose of this paper is rather to "understand the uncertainties" in the sense of characterizing them in a consistent manner based on the satellite data products available to the users, providing thus a solid basis for further investigations both on the model evaluation side and retrieval algorithm research field. To illustrate this the first sentence in the abstract has been changed to "To better understand and characterize current uncertainties…"

Philosophical:  the major purpose of this paper is to understand the usefulness of satellite remote sensing datasets for model evaluation. In that sense it is a different paper than some other studies that intercompared satellite datasets to find a single optimal dataset (and possibly understand the errors in others). In contrast, we want to understand how the ensemble of datasets behaves. Interestingly, we find that while long-term errors vs

AERONET can differ quite a bit from site to site, on the whole many datasets yield a similar performance (Fig. 7, 8 or 23).

We also want to add that Fig. 23 (Fig. 30 before) was remade as we realized only AERONET sites in land grid-boxes were used. The new figure uses all available AERONET sites (it increases the number of used sites by ~ 15%). This only affects the figure slightly and does not change our results.

Specific comments

On page 4 the authors state: "We will provide evidence that cloud masking is the dominant factor....". Please be more specific about where in the paper such evidence is provided.

A good suggestion, we now provide a brief listing of such evidence. In the summary we now say: "The evidence consist of the following observations: 1) biases vs AERONET decrease with increasing coverage; 2) correlations with AERONET increase with increasing coverage; 3) satellite differences decrease with increasing coverage. The simplest explanation (Ockam's razor) would be that coverage is the complement of cloud fraction and as coverage goes down, cloud fraction (and cloud contamination) goes up."

On page 5 the authors state: "Most ocean boxes with observations will be in coastal regions, with some over isolated islands." Please explain the reason for this restriction.

The MAN web-site indicates that data are obtained by Mircrotops deployed on ships all over the world. Why are those data not used in this study?

This statement refers to AERONET observations and is consequently trivial. The original text was misleading and has been corrected. MAN data over 'deep' ocean are used just as well as MAN data from coastal areas.

On page 6, the following sentence appears: "Bootstrapping has been shown to be reliable even for relatively small sample sizes." A reference in support of this statement seems to be required.

Agreed, reference added. Bootstrap Methods: A Guide for Practitioners and Researchers by Chernick discusses the issues of small sample sizes in detail and suggests that already for n>10 reasonable results may be expected.

on page 9 the authors remark: "It is not surprising that some products will have better cloud masking than others ...". Some explanation would be useful here.

Agreed, some explanation is now added. In short: different products' cloud masks are based on different sort of raw observations (e.g. spatial resolution, wavelength bands). Not all such observations are equally suited to cloud masking. In addition, there is always some

freedom in setting thresholds, depending on whether one wants to have as many aerosol observations as possible or as strict a cloud masking as possible.

On page 10 the authors write: "Terra/Aqua-DT, by the way, sometimes produces neg- ative AOD leading to e.g. very low values for averaged AOD over Australia." What is the reason for this problem?

Algorithmic. Overestimation of surface albedo may lead to negative AOD. Some algorithms mask out such negative values but the DarkTarget team prefers to keep them to prevent skewing observations to larger AOD values. We have added the sentence "The DarkTarget algorithm can retrieve negative AOD values, e.g. as a result of overestimating surface albedo, and the DarkTarget team retains those values to prevent skewing the whole dataset to larger values".

On bottom of page 10: "The contrasts in the differences over land and neighbouring ocean (e.g. African outflow for Terra-DT with AATSR-SU or AATSR-ADV, or Aqua- DT with OMAERUV, or AVHRR with SeaWiFS) may likewise be driven by albedo treatment." Some more discussion of this "albedo treatment" issue would be useful.

As we said before, we don't attempt to interpret differences between algorithms. We have replaced the word "treatment' with 'estimate'.

On page 11 the authors write: "The three products based on the DeepBlue algorithm (Aqua- DB, AVHRR and SeaWiFS) suggest that already small algorithmic differences can yield significant differences." Please be more specific about what is meant by "small algorithmic differences".

SeaWiFS and especially AVHRR have less VIS channels than MODIS for which the algorithm was originally developed. This requires additional assumptions for the algorithm to work. In addition cloud masking works different, again due to different wavelength bands but also different pixel sizes. We have added additional explanation.

On page 12, the authors state: "Diversity is generally lowest over ocean, never reaching over 30% while over land values of 100% are possible." Please explain this result.

It is generally assumed that over ocean retrievals are more accurate because: 1) surface albedo is low; 2) scenes are more homogenous. The official MODIS DarkTarget uncertainty estimates over ocean and land support this. Also, over-ocean less datasets are available and they tend to have more in common. E.g. for afternoon datasets, only MODIS DarkTarget, SeaWiFS, AVHRR and OMAERUV provide data over majority of oceans. However, SeaWiFS and AVHRR use a similar algorithm (SOAR).

On page 13, the authors state: "We see that the diversity goes down when the mean AOD increases, and goes up when the uncertainty in cloud masking increases. This is as one would expect." Please elaborate on why this result is to be expected.

Mean AOD (i.e. averaged over all datasets) should be a reasonable estimate of true AOD (note we do not say it is the optimal estimate). Higher AOD should make it easier to perform

aerosol retrievals more reliably and should result in lower diversity. Large uncertainty in cloud masking implies that at least some products suffer from cloud contamination and diversity will be larger. We have added more explanation in the paper.

Technical corrections:

We have changed all instances of the word 'data' to be treated as plural.

Reply to reviewer 2

We would like to thank anonymous reviewer 2 for their time to review our paper and provide useful comments to improve the text.

General comments

[..] I had a chance to read the other review comment, and I am in general agreement with the comments there. I strongly agree that there could be brief descriptions for each individual AOD algorithms, and how each one of them treats clouds and surface. This information could be put in an appendix. [..]

The reviewer asks for more details on the algorithmic procedures used in deriving fourteen different products (11, if we count Aqua-Terra as single products). While this request is understandable, we choose not to include this information for several reasons:

Practical: it would significantly increase the size of the paper which is already large. Note that we provide references to the papers that describe the individual products and their algorithms.

Scientific: such a description might make sense if we would consequently be able to use it to interpret results. However, that is not the purpose of this paper. We expect it would not be easy to make such an interpretation (see e.g. Holzer-Popp et al. 2013). We believe that the purpose of this paper is rather to "understand the uncertainties" in the sense of characterizing them in a consistent manner based on the satellite data products available to the users, providing thus a solid basis for further investigations both on the model evaluation side and retrieval algorithm research field. To illustrate this the first sentence in the abstract has been changed to "To better understand and characterize current uncertainties…"

Philosophical: the purpose of this paper is to understand the usefulness of satellite remote sensing datasets for model evaluation. In that sense it is a different paper than some other studies that intercompared satellite datasets to find a single optimal dataset (and possibly understand the errors in others). In contrast, we want to understand how the ensemble of datasets behaves. Interestingly, we find that while long-term errors vs AERONET can differ quite a bit from site to site, on the whole many datasets yield a similar performance (Fig. 12).

[..] The authors look only at global scale. However it is expected that the importance of surface albedo may show up in some regions, e.g., mountainous regions. So it may be worth some regional analysis on the impact of spatial coverage upon evaluation result.

As mentioned in Section 3.3, we habitually performed our analyses for individual regions as well but chose not to show those results (no significant deviations from the global analysis were found). In the case the reviewer is referring to we do *not* argue that surface albedo has

no impact on product error vs AERONET. Rather we argue that the observed behavior (better product performance at higher spatial coverage) is hard to explain using surface albedo.

In addition, the detailed analysis is acknowledged, however a total number of 30 for figures is relatively high. Authors could consider moving some figures into supplement, and making the most important results stand out in the manuscript.

A good idea. We have put several figures in a supplement. In particular, several figures relating to the selection of collocation criteria and site selection as well as the MAN analysis were moved to the supplement.

We also want to add that Fig. 23 (Fig. 30 before) was remade as we realized only AERONET sites in land grid-boxes were used. The new figure uses all available AERONET sites (it increases the number of used sites by ~ 15%). This only affects the figure slightly and does not change our results.

Minor points

It is noted that the author has his own writing style, which is fluent, however, not necessarily formal. For example, the second sentence on Page 13 line 15, starts with "E.g." which should be "For example. . .." And there are many more places which are not listed in this review. I would leave to the editor if minor English editing is required.

That instance has been corrected. We have corrected a few other grammatical errors as well.

P1 Line 10: It is confusing what "spatial coverage" means here. Please be specific.

It is an estimate of the fraction of a 1° by 1° grid-box covered by L2 AOD retrievals, at a certain time. See page 4, line 14. It is an estimate, as it is difficult to properly account for pixel distortions at higher viewing zenith angles.

P2 Line 25: "AOD (Aerosol Optical Depth)" should be "Aerosol Optical Depth (AOD)", ie., full expression first, and abbreviation next. Same thing for Line 28, MODIS, MISR abbreviations, and AERONET.

Corrected.

P3 Line 8, please define "super-observation". P4 line 27-28 this sentence reads awkward.

We have added an appendix that describes in more detail the construction of our datasets.

P6 Line 18-19, I don't think this averaging over all sites of their bias and correlation is a novel error metric.

We have no knowledge of papers that use such metric. It certainly is not a common metric and measures something different than the common global bias and correlation. Although

the reviewer provides no indication where this metric has been used before, we have removed the adjective 'novel'.

P23 Table 1, Under "Spatial" resolution column , there is a "?" for Kinne (2009), which needs to be fulfilled.

Corrected.

Table 2, It would be nice to provide information about time span of each product.

In this study, we use 2006, 2008 and 2010 but the products extend over many more years.

Figure 5. There seem to be missing panels based on the figure caption. The figure only shows evaluation result with collocated AERONET observation within 3hours, but result with AERONET observation within 1hour is also expected.

The figure shows performance results (bias, correlation, RMSD) for data collocated within 3 hours (vertical axis) *vs* 1 hour (horizontal axis). No figures are missing. We have improved the caption to prevent this misunderstanding.

Figure 8. Colors representing different satellite products overlap each other. For about half of the satellite products, it is impossible to see their presence. Please think of different plotting method (e.g., making hatching less dense, with different patterns,, smaller area on top of larger area) so that large area does not totally cover smaller areas, etc,) to make all the products visually identifiable.

We understand the problem but would argue it is not that important; the main message here is that there is uncertainty in these analyses that precludes hard conclusions on optimal dataset. We have moved this figure to the supplement to create more space anyway.

P10 Line 10 To be consistent with the rest of the manuscript, remove "FMI" in "AATSR- FMI-ADV".

Corrected.

Figure 24. This figure gives the ratio of difference between satellite AOD products for spatial coverage at 90-100% to 0-10%, which corresponds to approximately 0-10% to 90-100% cloud coverage if cloud is considered the largest impactor for the AOD spatial coverage. It would be nice to break up into a few similar panels, e.g, similar subplots with relatively low, median and high spatial coverages, e.g. around 10%, 30% , 50%, 70%, 90%. This information would be useful for AOD data assimilation users, as cloud fraction is one of the used information (as threshold) of AOD data to generate DA- quality product for aerosol DA. This would give some guidance on what could fraction is reasonable to obtain AOD consistency among multiple satellite products in AOD DA efforts.

A good idea. We will include this figure in a supplement. Our analysis shows that even at fairly high spatial coverage(~ 50%), there is only a modest ~25% reduction in AOD difference

between satellite pairs compared to 0-10% coverage. That is not surprising since our original analysis suggested no more than a 40% reduction in difference for coverages of ~ 95%.

Figure 27. What do the contours over north Africa, Arabian Peninsula and Siberia represent? This is explained in the text, but it would be nice to describe in the figure caption also.

Agreed.

Page 13 Line 8, "de average. . ." typo?

Corrected.

Figure 30 caption, typo "diveisity"

Corrected.

Page 14, Line 19. Summary section, ". . .. . ..MISR because the product was in the mid- dle of an update cycle, and VIRRS because it was only launched in 2011." I understand the meaning of this sentence, but formal English is preferred as this is for publication. Also I believe there is a typo for VIIRS.

Corrected.

Page 14, line 20. "For MODIS and AATSR, four resp. three different retrieval algorithms were used". See comment above.

Corrected.

Page15, Line 31, "patters", typo.

Corrected.